# *Drosophila* model to clarify the pathological significance of OPA1 in autosomal dominant optic atrophy

**Yohei Nitta[1†], Jiro Osaka[1,2†], Ryuto Maki[2], Satoko Hakeda-Suzuki[2,3], Emiko Suzuki[4,5], Satoshi Ueki[6], Takashi Suzuki[2], Atsushi Sugie[1*]**

[1]Brain Research Institute, Niigata University, Niigata, Japan; [2]School of Life Science and Technology, Tokyo Institute of Technology, Yokohama, Japan; [3]Research Initiatives and Promotion Organization, Yokohama National University, Yokohama, Japan; [4]Department of Biological Sciences, Graduate School of Science, Tokyo Metropolitan University, Hachioji, Japan; [5]Department of Gene Function and Phenomics, National Institute of Genetics, Mishima, Japan; [6]Division of Ophthalmology and Visual Science, Graduate School of Medical and Dental Sciences, Niigata University, Niigata, Japan

**\*For correspondence:**
atsushi.sugie@bri.niigata-u.ac.jp

[†]These authors contributed equally to this work

**Competing interest:** The authors declare that no competing interests exist.

**Abstract** Autosomal dominant optic atrophy (DOA) is a progressive form of blindness caused by degeneration of retinal ganglion cells and their axons, mainly caused by mutations in the OPA1 mitochondrial dynamin *like* GTPase (*OPA1*) gene. *OPA1* encodes a dynamin-like GTPase present in the mitochondrial inner membrane. When associated with OPA1 mutations, DOA can present not only ocular symptoms but also multi-organ symptoms (DOA plus). DOA plus often results from point mutations in the GTPase domain, which are assumed to have dominant-negative effects. However, the presence of mutations in the GTPase domain does not always result in DOA plus. Therefore, an experimental system to distinguish between DOA and DOA plus is needed. In this study, we found that loss-of-function mutations of the *dOPA1* gene in *Drosophila* can imitate the pathology of optic nerve degeneration observed in DOA. We successfully rescued this degeneration by expressing the human *OPA1* (*hOPA1*) gene, indicating that *hOPA1* is functionally interchangeable with *dOPA1* in the fly system. However, mutations previously identified did not ameliorate the *dOPA1* deficiency phenotype. By expressing both WT and DOA plus mutant *hOPA1* forms in the optic nerve of *dOPA1* mutants, we observed that DOA plus mutations suppressed the rescue, facilitating the distinction between loss-of-function and dominant-negative mutations in *hOPA1*. This fly model aids in distinguishing DOA from DOA plus and guides initial *hOPA1* mutation treatment strategies.

## eLife assessment

This study provides **valuable** insights into the complex genetics of dominant optic atrophy. Leveraging a fly model, the investigators provide **solid** evidence, albeit with small effect sizes, for a dominant negative mechanism of certain pathogenic variants that tend to cause more severe phenotypes, a long held hypothesis in the field. The work is of high interest to those in the optic atrophy and degeneration fields.

## Introduction

Autosomal dominant optic atrophy (DOA) is a progressive form of blindness characterized by selective degeneration of retinal ganglion cells (RGCs) and the axons that form the optic nerve. Despite

being a rare disease with a frequency of 1/12,000 to 1/50,000, DOA is the most commonly diagnosed form of hereditary optic neuropathy. In 2000, *OPA1 mitochondrial dynamin like GTPase* (*OPA1*) was identified as the causative gene for DOA (*Alexander et al., 2000*; *Delettre et al., 2000*).

*OPA1* encodes a dynamin-like GTPase located in the inner mitochondrial membrane (*Olichon et al., 2002*). It has various functions, including mitochondrial fusion, mitochondrial DNA (mtDNA) maintenance, control of cell death, and resistance to reactive oxygen species (ROS) (*Lenaers et al., 2021*). The most common pathogenic mutation in *OPA1* is the c.2708–2711 delTTAG deletion (*Toomes et al., 2001*). Five pathogenic mutations, including the one above, result in a substantial decrease in OPA1 protein in the fibroblasts from affected patients (*Zanna et al., 2008*), supporting the theory that DOA is caused by haploinsufficiency.

Optic atrophy is a characteristic feature of *OPA1*-associated DOA; however, multisystem symptoms have been reported in up to 20% of *OPA1* mutation carriers (*Yu-Wai-Man et al., 2010a*). This is called DOA plus. DOA plus can include sensorineural deafness (*Amati-Bonneau et al., 2008*), multiple sclerosis-like illness (*Verny et al., 2008*; *Yu-Wai-Man et al., 2016*), parkinsonism and dementia (*Carelli et al., 2015*; *Lynch et al., 2017*), and cardiomyopathy (*Spiegel et al., 2016*). The R445H mutation is a well-known mutation associated with DOA plus (*Amati-Bonneau et al., 2008*; *Amati-Bonneau et al., 2003*; *Shimizu et al., 2003*). The proportion of mutated OPA1 locations differs between patients with DOA and DOA plus. Missense mutations in the dynamin-featured structure, especially the GTPase domain of *OPA1*, are more likely to cause severe symptoms compared with loss-of-function (LOF) mutations such as deletions or splice site mutations (*Yu-Wai-Man et al., 2010a*). This is probably due to the dominant-negative (DN) effect of a DOA plus mutation on the normal *OPA1* allele, since OPA1 functions as a homo-oligomer (*Frezza et al., 2006*; *Olichon et al., 2006*); in fact, its yeast ortholog, Mgm1, forms oligomers, increasing the GTPase activity (*Meglei and McQuibban, 2009*; *Rujiviphat et al., 2009*). Particularly, the dynamin-featured structure accounts for 70% of the 233 pathogenic variants of DOA and DOA plus described in the locus-specific database (*Ferré et al., 2015*). However, mutations in these domains do not determine DOA plus, as it affects only 20% of all *OPA1* mutation carriers. Thus, experimental models to determine whether a gene mutation is LOF or DN are required to distinguish between DOA and DOA plus. Although currently there is no effective treatment for DOA or DOA plus, distinguishing between them allows earlier planning of interventions such as hearing aids, rather than relying solely on visual aids, necessary only for those with DOA plus. In the quest to differentiate between LOF and DN effects within the context of genetic mutations, precedents exist in simpler systems such as yeast and human fibroblasts. These models have provided valuable insights into the conserved functions of OPA1 across species, as evidenced by studies in yeast models (*Del Dotto et al., 2017*) and fibroblasts derived from patients harboring *OPA1* mutations (*Kane et al., 2017*). However, the ability to distinguish between LOF and DN effects in an in vivo model organism, particularly at the structural level of retinal axon degeneration, has remained elusive. This gap underscores the necessity for a more complex model that not only facilitates molecular analysis but also enables the examination of structural changes in axons and mitochondria, akin to those observed in the actual disease state.

As *OPA1* is conserved in various species, DOA models have been reported in vertebrates such as mice and zebrafish, simple organisms such as nematodes and fruit flies, and in vitro yeast and cultured cell models (*Del Dotto and Carelli, 2021*). In mouse models, focal RGC axons decrease in number with mitochondrial abnormalities at the electron microscope level, but the phenotype appears slowly, not allowing for analysis in a short time frame (*Alavi et al., 2007*; *Davies et al., 2007*; *Nguyen et al., 2011*; *Sarzi et al., 2012*). Zebrafish models have displayed developmental delays in eyes and heads, short length, and body axis abnormalities (*Eijkenboom et al., 2019*; *Rahn et al., 2013*) but have not been used to investigate optic nerve degeneration.

In *Caenorhabditis elegans*, most studies focused on muscle cells using mutants of *eat-3*, an *OPA1* homolog (*Kanazawa et al., 2008*; *Rolland et al., 2009*). However, the mitochondria in the posterior lateral microtubule neurons did not show any difference in size from its wild-type (WT) counterparts (*Byrne et al., 2019*). Thus, LOF of *eat-3* may have a limited impact on phenotypic outcomes in the nervous system. Nevertheless, a DOA model in which *hOPA1^K301A* is expressed in GABAergic motor neurons showed a decreased number of mitochondria in those cells (*Zaninello et al., 2020*). In a fly model, the *dOPA1* somatic clone in the eye exhibited a rough-eye phenotype, eye structural abnormalities, and increased MitoSOX fluorescence, which was partially rescued by vitamin E or SOD1

expression (*Yarosh et al., 2008*). Heterozygous *dOPA1* mutants showed shortened lifespan, elevated ROS levels, and irregular muscle tissue mitochondrial structures (*Tang et al., 2009*). Moreover, the electroretinogram pattern showed age-dependent decreases in the on-transient response, heart rate, negative geotaxis response, and increased heart arrhythmia due to heat shock stress (*Shahrestani et al., 2009*). However, except for mice, the evidence of optic nerve axonal abnormalities is limited in model organisms such as zebrafish, worms, and flies.

The structure of the *Drosophila* visual system is similar to that of mammals (*Sanes and Zipursky, 2010*), with conserved mechanisms of synaptogenesis and neural circuit formation (*Sanes and Zipursky, 2020*). *Drosophila* photoreceptors type R7/8 extend their retinal axons from the retina directly into the brain, forming synapses in the second optic ganglion medulla via the chiasma (*Figure 1A*), having potential application as a model for mammalian photoreceptors and RGCs. In this study, we defined the retinal axons of *Drosophila* as analogous to the human optic nerve. Based on this, we aimed to develop an experimental model for observing and quantifying degeneration in the *Drosophila* retinal axon (*Nitta et al., 2023*; *Richard et al., 2022*).

In this study, a *dOPA1* LOF could mimic the human DOA, in which mitochondrial fragmentation, increased ROS levels, and neurodegeneration occur in retinal axons. The axonal degeneration induced by the *dOPA1* mutant was rescued by the expression of human *OPA1* (*hOPA1*). This demonstrated that the function of *hOPA1* is the same as in the fly. Previously reported mutations failed to rescue the dOPA1 deficiency phenotype. To distinguish between the effect of LOF and DN mutations, we expressed the *hOPA1* gene of both WT and mutation in a genetic background in which *dOPA1* was deleted in retinal axons. The DOA plus mutations D438V and R445H inhibited the rescue by the WT of *hOPA1*. Taken together, we generated a new *Drosophila* DOA model which served to isolate the LOF or DN effect of *hOPA1* mutations.

## Results

### *dOPA1* depletion caused mitochondrial fragmentation in axon terminals

The OPA1 protein is located in the inner mitochondrial membrane. To confirm whether dOPA1 is also localized in the mitochondria of *Drosophila* retinal axons, we used a Gal4/UAS system (*Brand and Perrimon, 1993*) and expressed HA-tagged dOPA1 proteins using the eye-specific *GMR-Gal4* driver (*Newsome et al., 2000a*; *Newsome et al., 2000b*). The outer membrane of the mitochondria was visualized with mCherry-mito, which tagged the transmembrane domain of Miro with mCherry (*Vagnoni and Bullock, 2016*). Confocal microscopy revealed that the HA signals colocalized with mCherry-mito in the axon terminals of photoreceptor cells (*Figure 1B*). This indicated that dOPA1 colocalizes with mitochondria in retinal axons.

To examine the impact of *dOPA1* on mitochondrial structure and density, we analyzed mitochondria in *dOPA1* RNAi flies. The intensity of the mitochondrial signal was significantly reduced in the retinal axons of these flies (*Figure 1C and D*; quantification in *Figure 1E*). Previously, impaired mitochondrial transport was reported in the LOF of *marf* (MFN 1/2 homolog), which is necessary for the fusion of the outer membrane of mitochondria (*Sandoval et al., 2014*). Thus, we investigated if the LOF of *dOPA1* may cause transport defects, which may be responsible for the reduced mitochondrial density in retinal axons in *dOPA1* RNAi flies. To this end, we observed the photoreceptor neurons after puparium formation at 24 hr, which allowed us to simultaneously visualize the cell bodies and axon terminals of the photoreceptors (*Figure 1—figure supplement 1*). Both *dOPA1* knockdown and control flies showed mitochondrial signals in the axon terminals (*Figure 1—figure supplement 1*). However, no mitochondrial signals were observed in the axon terminals of flies with a knockdown of *milton*, which is thought to act as an adaptor between kinesin to mitochondria, promoting antero-grade transport (*Figure 1—figure supplement 1*). These findings suggest that trafficking defects are not the main cause of the mitochondrial mislocalization observed in the retinal axons of the *dOPA1* knockdown.

To further analyze the mitochondrial morphology of retinal axons in the LOF of *dOPA1*, we conducted an electron microscopy analysis (*Figure 1F and G*). Although *Drosophila* has eight types of photoreceptors, the axons of R1–6, which project to the lamina of the first optic lobe, can be identified without markers; thus, we chose them for this experiment. The *dOPA1* knockdown led to a significant decrease in mitochondrial size in R1–6 axons (*Figure 1F–H*). In addition, there was significant

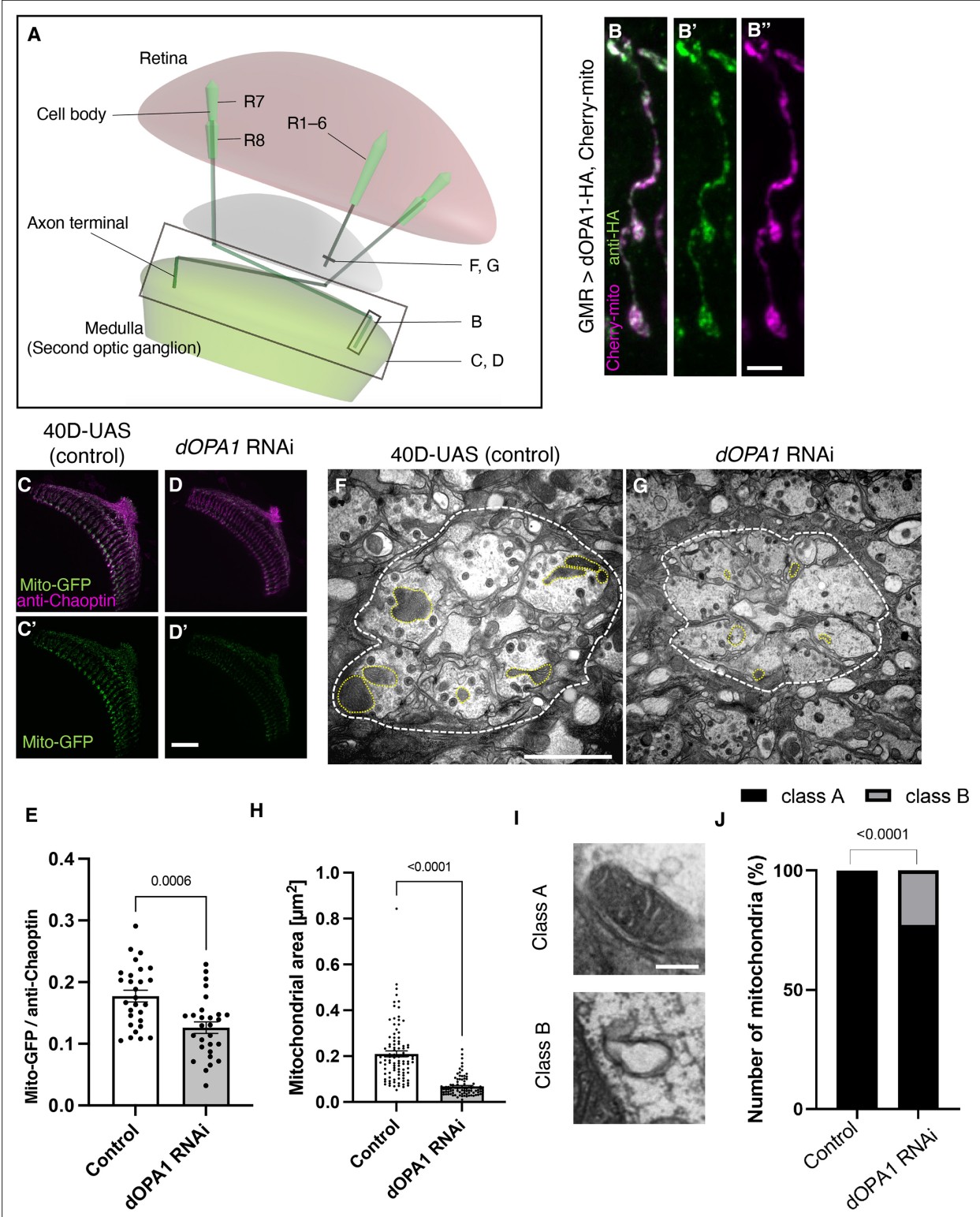

**Figure 1.** Effect of *dOPA1* knockdown on mitochondrial density and size in retinal axons. (**A**) Schematic illustration of the *Drosophila* visual system from a dorsal view, highlighting the arrangement of photoreceptor neurons. The rectangular area indicated in (B) represents a single R7/8 photoreceptor axon, as illustrated in the figure for (B). The rectangular areas highlighted in (C) and (D) encompass the entirety of R7/8 photoreceptor axons, as shown in the figures for (C) and (D). The lines marked in (F) and (G) denote cross-sections of R1–6 photoreceptor axons, as depicted in the figures for (F) and (G). (**B**) Visualization of mitochondria and dOPA1 in a set of R7/8 axons. Mitochondria were visualized using the expression of *mCherry-mito* driven by *GMR-Gal4* (magenta) while HA-tagged dOPA1 was immunostained using the anti-HA antibody (green). Scale bar: 3 µm. (**C, D**) Representations

*Figure 1 continued on next page*

*Figure 1 continued*

of mitochondria visualized using the expression of *Mito-GFP* driven by *GMR-Gal4* (green), 1 day after eclosion. Retinal axons were labeled using the anti-Chaoptin antibody (magenta). Scale bar: 30 µm. (**E**) Quantification of Mito-GFP intensity. Mito-GFP levels were calculated by dividing the Mito-GFP fluorescence intensity by the anti-Chaoptin signal intensity. Control (n=28 optic lobes) and *dOPA1* RNAi (n=28 optic lobes). The data are presented as mean ± SEM. (**F, G**) Electron micrographs of cross-sections of the R1–6 retinal axons in the lamina of the Control (**F**) and d*OPA1* knockdown (**G**) on the day of eclosion. Yellow circles and white dotted lines indicate mitochondria and a lamina column, respectively. Scale bar: 2 µm. (**H**) Quantification of the mitochondrial area. Control (n=96 mitochondria) and *dOPA1* RNAi (n=92 mitochondria). Data are expressed as mean ± SEM. (**I**) Representative EM images showing mitochondria with a densely packed matrix structure (classified as Class A) and collapsed mitochondria (classified as Class B). Scale bar: 500 nm. (**J**) Quantification of the mitochondria classified into Class A and Class B. Control (n=96 mitochondria) and *dOPA1* RNAi (n=92 mitochondria). See the *Supplementary file 1* for the genotypes of the *Drosophila* used in the experiments.

The online version of this article includes the following figure supplement(s) for figure 1:

**Figure supplement 1.** Analysis of trafficking defects in axons of *dOPA1* knockdown flies.

heterogeneity in healthy mitochondria, ranging from dense matrix (Class A) to swollen mitochondria with a hypodense matrix (Class B) (*Figure 1I*). Knocking down *dOpa1* in R1–6 axon terminals resulted in severely abnormal mitochondrial ultrastructure (22.8%; *Figure 1J*). These results suggest that *dOPA1* is involved in maintaining proper mitochondrial morphology by promoting mitochondrial fusion in retinal axons.

To verify whether ROS is increased in the retinal axons in *dOPA1* knockdown photoreceptors, we measured the ROS levels in the knockdown retinal axons using MitoSOX, a mitochondrial superoxide indicator (*Yarosh et al., 2008*). The results showed significantly elevated ROS levels in *dOPA1* knockdown retinal axons compared to control (*Figure 2A and B*; quantification in *Figure 2C*).

Given that an increase in mitophagy activity has been reported in mouse RGCs and nematode ADOA models (*Zaninello et al., 2022*; *Zaninello et al., 2020*), the mitoQC marker, an established indicator of mitophagy activity, was expressed in the photoreceptors of *Drosophila*. The mitoQC reporter consists of a tandem mCherry-GFP tag that localizes to the outer membrane of mitochondria (*Lee et al., 2018*). This construct allows the measurement of mitophagy by detecting an increase in the red-only mCherry signal when the GFP is degraded after mitochondria are transported to lysosomes. Post *dOPA1* knockdown, we observed a significant elevation in mCherry-positive and GFP-negative puncta signals at 1 week, demonstrating an activation of mitophagy as a consequence of *dOPA1* knockdown (*Figure 2D–H*).

## The *dOPA1* LOF leads to progressive distal degeneration of *Drosophila* photoreceptors

We tested whether knocking down *dOPA1* causes optic nerve degeneration in *Drosophila*. To this end, we performed an RNAi experiment using the *GMR-Gal4* driver and evaluated the number of retinal axons projecting to the second optic ganglion medulla (*Figure 3A*). To compare the number of retinal axon terminals – R7 axons – between genotypes, we used a previously developed automated method, MeDUsA (**me**thod for the quantification of **d**egeneration **us**ing fly **a**xons) (*Nitta et al., 2023*). We have assessed the extent of their reduction from the total axonal terminal count, thereby determining the degree of axonal terminal degeneration (*Nitta et al., 2023*; *Richard et al., 2022*). Our results showed that the *dOPA1* knockdown caused axonal degeneration 1 day after eclosion, further decreasing 1 week later (*Figure 3B–E*, quantification in *Figure 3F*). We obtained similar results with an independent alternative RNAi line (*Figure 3—figure supplement 1*). To determine whether the degeneration was limited to the axons, we counted the number of rhabdomeres in the cell bodies. Each compound eye comprises 700–800 ommatidia, with each ommatidium containing eight types of photoreceptors, designated as R1–R8. The distal region of an ommatidium reveals the R1–R7 photoreceptors with a stereotypical arrangement of rhabdomeres (arrowheads in *Figure 3G–J*). Since it is easy to evaluate retinal degeneration by counting the number of rhabdomeres, a decrease in their number indicates degeneration of the photoreceptor cell body. The *dOPA1* knockdown caused a significant decrease in the number of rhabdomeres per ommatidium in 1-week-old adults (*Figure 3G–J*, quantification in *Figure 3K*). Interestingly, although the retinal axons had already decreased on the day of eclosion, the ommatidia remained intact (*Figure 3F and K*). Note that the ommatidia in R7 remained intact after 1 week (*Figure 3J*). Our results indicate that the *dOPA1* knockdown causes degeneration in both axonal terminals and cell bodies, with the neuronal degeneration starting from the axons and continuing in

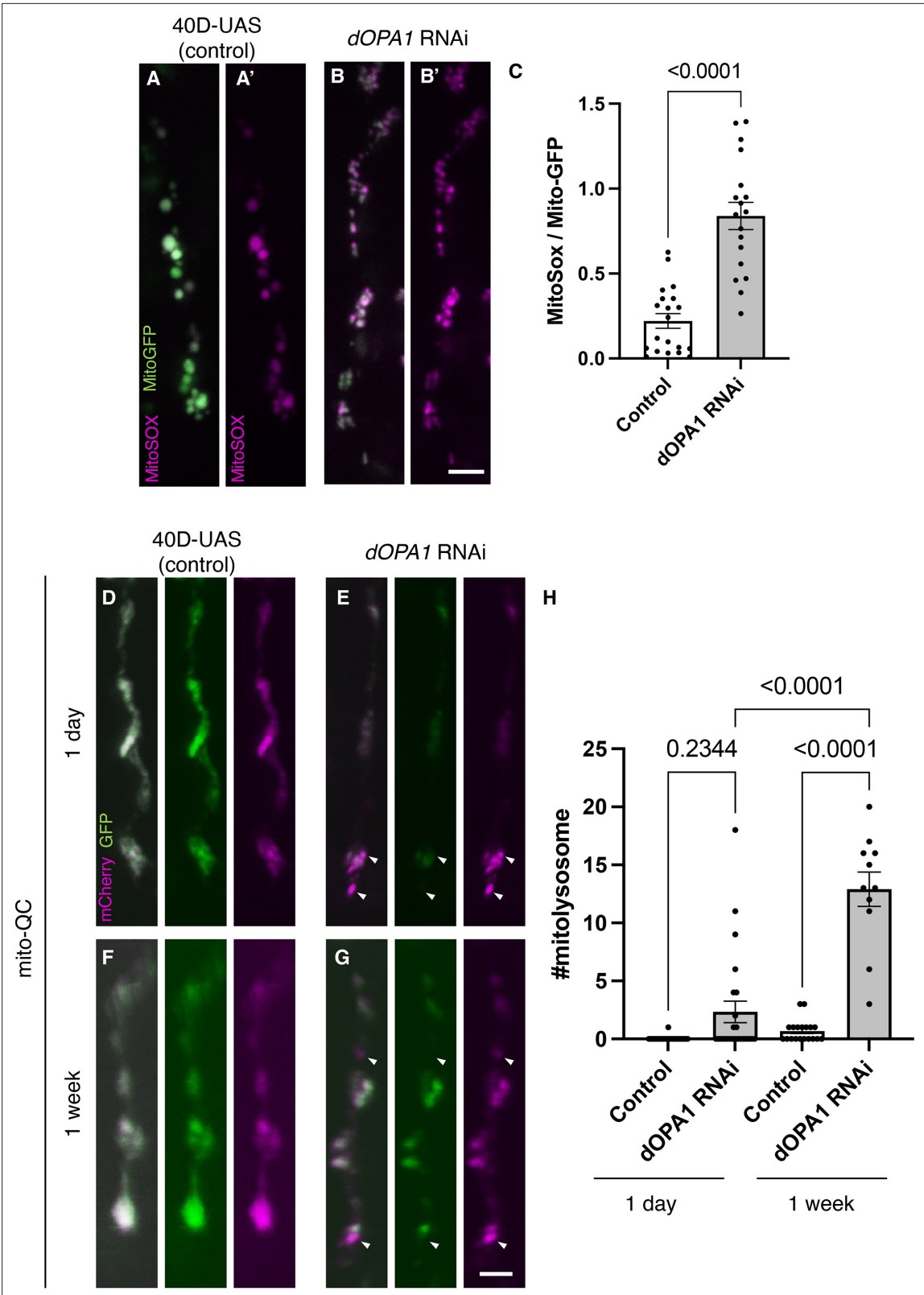

**Figure 2.** Elevated reactive oxygen species (ROS) and mitophagy activity levels in the *dOPA1* knockdown. (**A, B**) Mitochondrial ROS levels in a set of the R7/8 retinal axon visualized using Mito-GFP (green) driven by *GMR-Gal4* and MitoSOX (magenta) to specifically detect superoxide, an indicator of oxidative stress. Representative images of a retinal axon in the Control (**A, A'**) and *dOPA1* knockdown (**B, B'**). (**C**) Quantification of ROS levels for each genotype. ROS levels were determined by dividing the fluorescence intensity of MitoSOX by that of Mito-GFP, highlighting the significant increase

*Figure 2 continued on next page*

*Figure 2 continued*

in ROS levels in *dOPA1* knockdown flies compared to controls, suggesting enhanced mitochondrial stress and dysfunction upon *dOPA1* depletion. Control (n=20 optic lobes) and *dOPA1* RNAi (n=18 optic lobes). Scale bar: 3 µm. Data are presented as mean ± SEM. (**D, E**) Mitophagy activity in a set of the R7/8 retinal axon visualized by a genetic marker, mitoQC, driven by *GMR-Gal4*. Representative images of a retinal axon in the Control 1 day after eclosion (**D**), *dOPA1* knockdown 1 day after eclosion (**E**), the Control 1 week after eclosion (**F**), *dOPA1* knockdown 1 week after eclosion (**G**). The arrowheads indicate a mitolysosome that is positive for mCherry and negative for GFP. (**H**) Quantification of the mitophagy activity each genotype. The number of the mitolysosomes were counted in each genotype. The significant increase in the number of the mitolysosomes in *dOPA1* knockdown flies compared to controls after 1 week, suggesting enhanced mitophagy upon *dOPA1* depletion. Control 1d (n=18 optic lobes), Control 1 week (n=19 optic lobe), *dOPA1* RNAi 1d (n=24 optic lobes), and *dOPA1* RNAi 1 week (n=11 optic lobes). Scale bar: 3 µm. Data are presented as mean ± SEM. See the **Supplementary file 1** for the genotypes of the *Drosophila* used in the experiments.

the cell bodies. A previous report indicated that the compound eyes of homozygous mutations of *dOPA1* displayed a glossy eye phenotype (*Yarosh et al., 2008*). Upon knocking down *dOPA1* using the *GMR-Gal4* driver, we also observed a glossy eye-like rough-eye phenotype in the compound eyes (*Figure 3—figure supplement 2*). The *GMR-Gal4* driver does not exclusively target Gal4 expression to photoreceptor cells. Consequently, the observed retinal axonal degeneration could potentially be secondary to abnormalities in support cells external to the photoreceptors.

To test whether our RNAi results reflect *dOPA1* downregulation, we performed mutant analysis of the *dOPA1* gene in retinal axons. For this, we used the *dOPA1^{s3475}* hypomorphic mutant allele, which has a P-element insertion in exon 2 (*Yarosh et al., 2008*). Somatic mosaic clones were generated in the retinal axons using the FLP/FRT system (*Golic and Lindquist, 1989*) because homozygous mutant alleles are embryonic lethal (*Sandoval et al., 2014*; *Yarosh et al., 2008*). Note that the mutant clone analysis was conducted in a context where mutant and heterozygous cells coexist as a mosaic, and it was not possible to distinguish between them. Using MeDUsA, we found a significantly lower number of *dOPA1* mutant retinal axons than in controls 1 day after eclosion (*Figure 4A and B*, quantification in *Figure 4D*). To determine if *dOPA1* is responsible for axon neurodegeneration, we quantify the number of the axons in the *dOPA1* eye clone fly with the expression of *dOPA1* at 1 day after eclosion and found that *dOPA1* expression partially rescued the axonal degeneration (*Figure 4C*, quantification in *Figure 4D*). Our results indicate that *dOPA1* is necessary for maintaining the number of retinal axons in the *Drosophila* visual system. In conclusion, the neurodegeneration observed in the LOF of *dOPA1* is due to the progressive loss of retinal axons, as well as the mammal optic nerve.

## Investigating disease mutations in OPA1 using a fly DOA model to confirm their pathological significance

Our observations in *dOPA1* LOF flies confirmed mitochondrial fragmentation, increased ROS levels, and degeneration of retinal axons, as reported in mammals. These data indicated that the function of *OPA1* may be conserved across *Drosophila* and humans. Thus, we tested whether *hOPA1* could effectively replace *dOPA1* in flies by generating a transgenic organism to express *hOPA1* in *Drosophila*. Following the UAS sequence, that the yeast-derived transcription factor Gal4 binds, the construct included an HA tag, followed by the *hOPA1* gene, and a myc tag (*Figure 5A*). This construct was expressed in *Drosophila* and protein expression was confirmed by western blotting. The HA tag, attached to the N-terminus, was immunoblotted resulting in a band of the expected full-length size (*Figure 5B*). For the myc tag at the C-terminus, a strong band and two weak bands were observed at the upper and lower positions (*Figure 5B*). Comparison of the expression levels across the variants revealed no significant differences in protein expression (*Figure 5C*). We also employed a human anti-OPA1 antibody to verify the expression of hOPA1 in vivo in *Drosophila*, detecting both the long and short forms of OPA1, band 2 and band 4, respectively (*Figure 5D*).

Using this *UAS-hOPA1*, we performed rescue experiments. *hOPA1* expression in the retinal axons of the *dOPA1* mutant, which generated somatic cell clones, partially rescued the number of axons (*Figure 5E*). These results suggest that *hOPA1* and *dOPA1* are interchangeable. We also generated and expressed the *hOPA1* mutation 2708-2711del, which is known to cause DOA, the I382M mutation, located in the GTPase domain and associated with DOA, as well as the D438V and R445H mutations, also located in the GTPase domain and associated with DOA plus. The expression of these mutations in retinal axons failed to rescue the *dOPA1* deficiency to the same extent as the WT *hOPA1* (*Figure 5E*). Importantly, unlike the D438V and R445H mutations, the 2708-2711del and

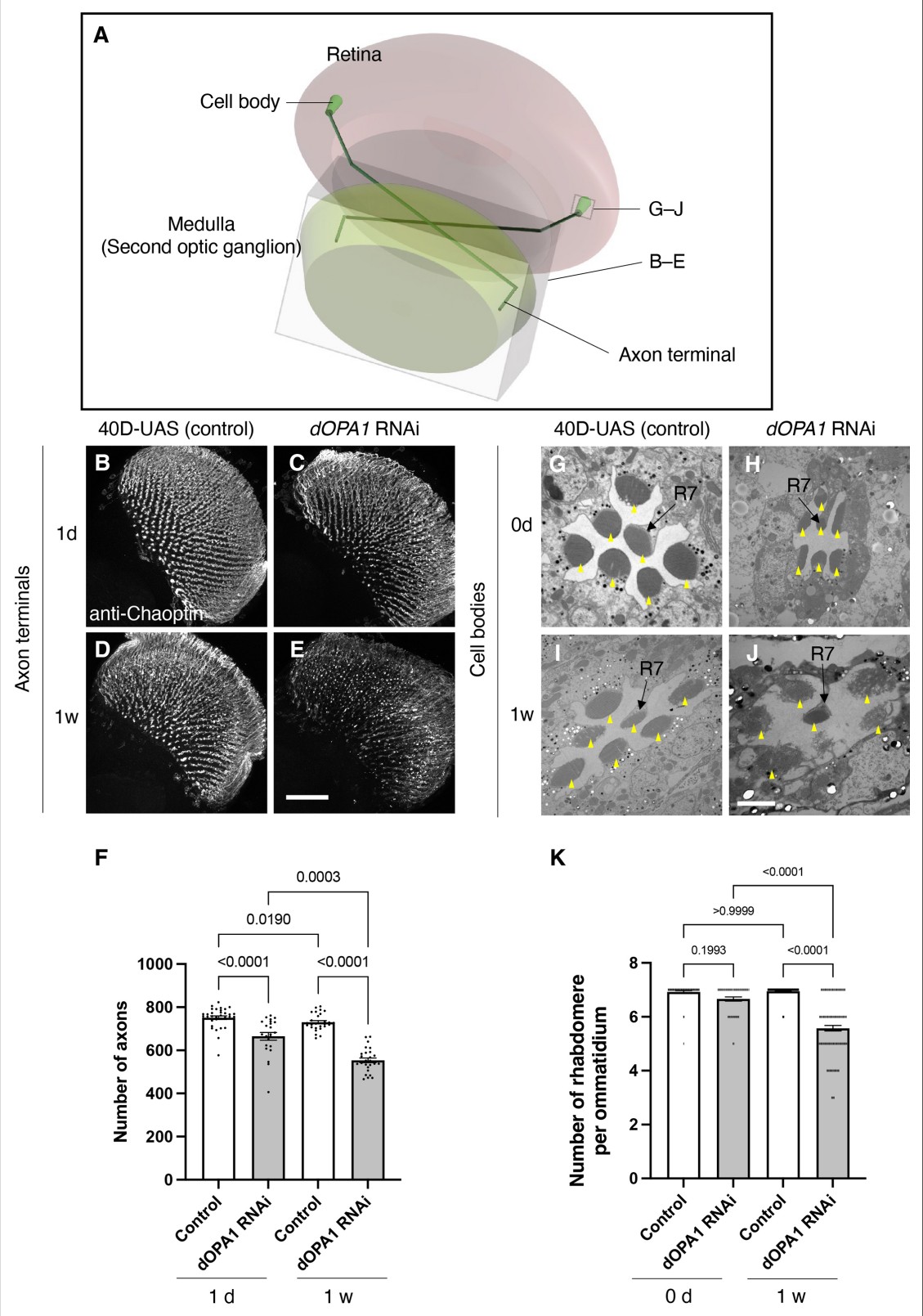

**Figure 3.** Effect of *dOPA1* knockdown on photoreceptor neurodegeneration in *Drosophila*. Schematic of the visual system in *Drosophila*. A set of R7/8 photoreceptors project their axons to the second optic ganglion medulla. The rectangular prism indicated in B–E represents the area of the whole medulla, as shown in the figure for (B–E). The square marked in G–J denote cross-sections of the cell bodies of the photoreceptors, as depicted in the figures for (G–J). (B–E) All R7 and R8 axon terminals project to the medulla. *dOPA1* in the photoreceptor was knocked down by *GMR-Gal4. 40D-*

*Figure 3 continued on next page*

*Figure 3 continued*

*UAS* was used as a Control to match the number of *UAS* sequences recognized by Gal4. One day after eclosion in Control (**B**) and *dOPA1* knockdown (**C**), and 1 week after in Control (**D**) and *dOPA1* knockdown (**E**). The retinal axons were stained with anti-Chaoptin, a photoreceptor-specific antibody. Scale bar = 50 μm. (**F**) Number of axons at each time point in R7 neurons for each situation quantified by MeDUsA (<u>me</u>thod for the quantification of degeneration <u>us</u>ing fly <u>a</u>xons). At 1 day of Control (n=34 optic lobes) and *dOPA1* RNAi (n=23 optic lobes), and at 1 week of Control (n=28 optic lobes) and *dOPA1* RNAi (n=26 optic lobes). Data are expressed as mean ± SEM. (**G–J**) Electron micrographs of cross-sections of the photoreceptor cell bodies in the ommatidia of the retina. Rhabdomeres are shown as yellow arrowheads. The day of eclosion in Control (**G**) and *dOPA1* knockdown (**H**), and 1-week-old adults of Control (**I**) and *dOPA1* knockdown (**J**). Scale bar = 2 μm. (**K**) Quantification of the number of rhabdomeres for each genotype and time point. At 0 day in Control (n=50 ommatidia) and *dOPA1* RNAi (n=127 ommatidia), and at 1 week in Control (n=50 ommatidia) and *dOPA1* RNAi (n=117 ommatidia). Data are expressed as mean ± SEM. See the ***Supplementary file 1*** for the genotypes of the *Drosophila* used in the experiments.

The online version of this article includes the following figure supplement(s) for figure 3:

**Figure supplement 1.** The impact on the retinal axons of another RNAi line in *Drosophila OPA1*.

**Figure supplement 2.** Evaluation of the impact on *Drosophila* compound eyes due to knockdown of *dOPA1*.

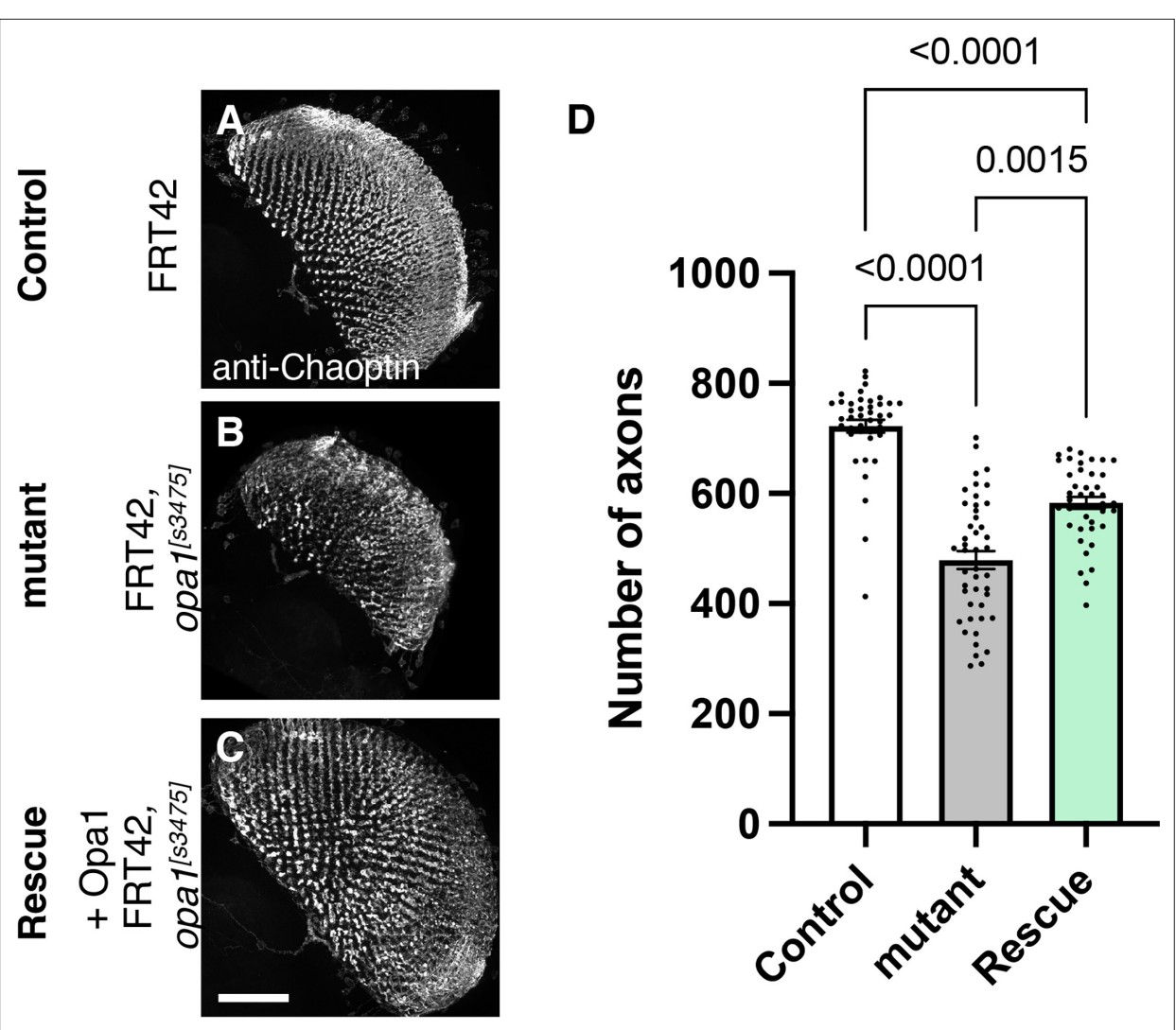

**Figure 4.** Role of *dOPA1* in the retinal axon according to mutant analysis. (A–C) Representations of retinal axons labeled with an anti-Chaoptin antibody (gray), 1 day after eclosion. The images show Control (**A**), *dOPA1^{s3475}* somatic mosaic flies (**B**), and *dOPA1^{s3475}* somatic mosaic flies expressing the eye-specific full-length *dOPA1* (**C**). Scale bar: 50 μm. (**D**) Quantification of the number of axons for each genotype. Control (n=41 optic lobes), mutant (n=45 optic lobes), and rescue (n=41 optic lobes). The data are presented as mean ± SEM. See the ***Supplementary file 1*** for the genotypes of the *Drosophila* used in the experiments.

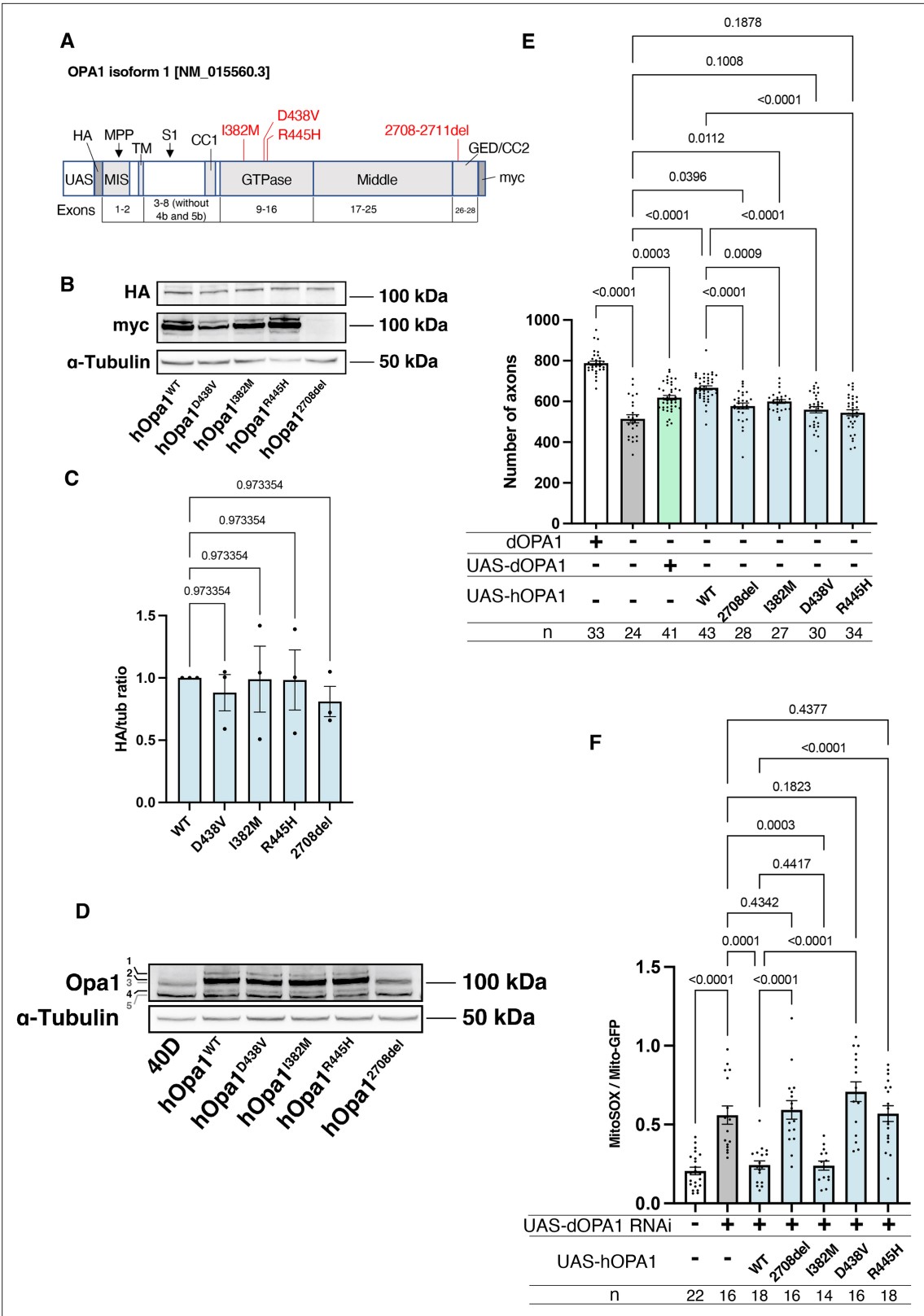

**Figure 5.** Verification of the pathological significance of disease mutations in *hOPA1*. (**A**) Schematic illustration of the *hOPA1* gene construct with HA and myc tags in the UAS-based vector. *hOPA1* includes a mitochondrial import sequence (MIS) cleaved by mitochondrial processing peptidase (MPP), a transmembrane region (TM), a coiled-coil region (CC1), a GTPase domain, a Middle domain, and a GTPase effector domain (GED) containing a coiled-coil region (CC2). The sites of variants (I382M, D438V, R445H, and a deletion from 2708 to 2711) are shown in red. S1 is the site cleaved

*Figure 5 continued on next page*

*Figure 5 continued*

by OMA1. (**B**) Western blot analysis to confirm the expression of hOPA1 variants. hOPA1_WT, hOPA1_D438V, hOPA1_I382M, hOPA1_R445H, and hOPA1_2708del were expressed in whole *Drosophila* bodies and detected using anti-HA and anti-myc antibodies. α-Tubulin was used as a loading Control. (**C**) Quantification of the expression levels of the OPA1 protein for each variant. hOPA1_WT (n=3), D438V (n=3), I382M (n=3), R445H (n=3), and 2708del (n=3). The data are presented as mean ± SEM. (**D**) Detection of hOPA1 expression in vivo in *Drosophila* using anti-OPA1. Band 1 represents the full-length OPA1, which includes the MIS, while Band 2 corresponds to the long-form OPA1 (L-OPA1), cleaved at the MPP. Bands 3 and 5 potentially detect the endogenous dOPA1. Band 4 is thought to represent the short-form OPA1 (S-OPA1). (**E**) Rescue experiments were conducted to assess the expression of each *hOPA1* variant, including *dOPA1*, in the retina axons of *dOPA1* mutant somatic clones. The sample size is indicated (n). Data are presented as mean ± SEM. (**F**) Quantification of mitochondrial reactive oxygen species (ROS) levels upon expression of hOPA1 variants in the context of dOPA1 knockdown. Control (n=22 optic lobes), *dOPA1* RNAi (n=16 optic lobes), *hOPA1_WT* with *dOPA1* RNAi (n=18 optic lobes), *hOPA1_2708-2711del* with *dOPA1* RNAi (n=16 optic lobes), *hOPA1_I382M* with *dOPA1* RNAi (n=14 optic lobes), *hOPA1_D438V* with *dOPA1* RNAi (n=16 optic lobes), and *hOPA1_R445H* with *dOPA1* RNAi (n=18 optic lobes). Data are presented as mean ± SEM. See the ***Supplementary file 1*** for the genotypes of the *Drosophila* used in the experiments.

The online version of this article includes the following figure supplement(s) for figure 5:

**Figure supplement 1.** Effects of *hOPA1* variants on COXII expression and autophagy in the context of *dOPA1* knockdown.

I382M mutations could be weakly rescued. This suggests that these mutations result in LOF, but may retain some residual activity. These findings are also in line with the observed severity of the associated diseases, DOA and DOA plus. The D438V and R445H mutations were not significantly rescued, unlike the 2708-2711del and I382M mutations. Our methodology distinctively facilitates the quantitative evaluation of LOF severity by comparing the rescue capabilities of various mutations. Notably, the 2708-2711del and I382M mutations demonstrated only partial rescue, indicative of a hypomorphic effect with residual activity. In contrast, the D438V and R445H mutations failed to show significant rescue, suggesting a more profound LOF. The correlation between the partial rescue by the 2708-2711del and I382M mutations and their classification as hypomorphic is significant. Moreover, the observed differences in rescue efficacy correspond to the clinical severities associated with these mutations, namely in DOA and DOA plus disorders. Thus, our results substantiate the model's ability to quantitatively discriminate among mutations based on their impact on protein functionality, providing an insightful measure of LOF magnitude.

Furthermore, we assessed the potential for rescuing ROS signals. Similar to its effect on axonal degeneration, WT hOPA1 effectively mitigated the phenotype, whereas the 2708-2711del, D438V, and R445H mutants did not (***Figure 5F***). Importantly, the I382M variant also reduced ROS levels comparably to the WT. These findings demonstrate that both axonal degeneration and the increase in ROS caused by dOPA1 downregulation can be effectively counteracted by hOPA1. Although I382M retains partial functionality, it acts as a relatively weak hypomorph in this experimental setup.

We also conducted western blot analyses using anti-COXII and anti-Atg8a antibodies to assess changes in mitochondrial quantity and autophagy activity following the knockdown of *dOPA1*. Mitochondrial protein levels, indicated by COXII quantification, were evaluated to verify mitochondrial content, and the ratio of Atg8a-1 to Atg8a-2 was used to measure autophagy activation. For these experiments, *Tub-Gal4* was employed to systemically knock down *dOPA1*. Considering the lethality of a whole-body *dOPA1* knockdown, *Tub-Gal80^{TS}* was utilized to repress the knockdown until eclosion by maintaining the flies at 20°C. After eclosion, we increased the temperature to 29°C for 2 weeks to induce the knockdown or expression of *hOPA1* variants. The results revealed no significant differences across the genotypes tested (***Figure 5—figure supplement 1***).

## Distinction between LOF and DN mutations in *hOPA1* linked to DOA or DOA plus

Currently, the *hOPA1* mutations that contribute to DOA plus have been primarily located in the GTPase domain of *hOPA1*. The role of *hOPA1* as a GTPase is facilitated by its ability to form a polymer via the GED domain and Middle domain (***Li et al., 2019***). This led to the hypothesis that mutations in the GTPase domain can interact with WT hOPA1 but cannot activate the GTPase activity to show a DN effect. However, it has been challenging to clarify whether these mutations are DN or LOF, as there is a significant number of LOF mutations in the GTPase domain. Given that *dOPA1* can be substituted with *hOPA1*, we suspected that a DN effect could also be achieved by expressing the D438V or R445H mutations in the WT background. However, the axons did not degenerate despite expression and

monitoring for 2 weeks in adult flies (*Figure 6A*). The results presented in *Figure 5C* indicate that there are no significant differences in the expression levels among the variants, suggesting that variations in expression levels do not influence the outcomes. The amino acid sequences of hOPA1 and dOPA1 are highly conserved (72% identity), particularly in the GTPase domain. The Middle and GED domains are also relatively well preserved, with 54% and 64% agreement, respectively (*Figure 6—figure supplement 1*). These results imply that, although *dOPA1* and *hOPA1* are functionally interchangeable, they may not interact with each other. Consequently, we expressed *hOPA1* in the photoreceptor depleted of *dOPA1* by somatic clone; under these conditions, we further expressed *hOPA1* WT, 2708-2711del, and mutations in the GTPase domains, including R445H, I382M, and D438V. This allowed *hOPA1* to interact with itself and to verify the DN effect. As a result, the expression of the WT, 2708del, and I382M did not result in any changes when rescued by *hOPA1* substitution (*Figure 6B*). However, the rescue was significantly suppressed for D438V and R445H, which are known to cause DN; thus, the DN effect could be replicated (*Figure 6B*). In conclusion, we established an experimental model that can separate LOF and DN, the pathological significance of *hOPA1* mutations. We investigated the impacts of DN mutations on mitochondrial oxidation levels, mitochondrial quantity, and autophagy activation levels; however, none of these parameters showed statistical significance (*Figure 6—figure supplement 2*).

## Discussion

In this study, we found that LOF of *dOPA1* in *Drosophila* can imitate human DOA, in which the optic nerve degenerates. We were successful in reversing this degeneration by expressing *hOPA1* (*Figure 6C*). However, we could not rescue any previously reported mutations known to cause either DOA or DOA plus except for I382M. In the context of rescuing the retinal axons of the *dOPA1* mutant by expressing the WT of *hOPA1*, it was observed that only the DOA plus mutations suppressed the rescue (*Figure 6C*). This allowed us to distinguish between LOF and DN mutations in *hOPA1*. The fly model developed in this study will allow investigating new *hOPA1* mutations for DOA or DOA plus. We have previously utilized MeDUsA to quantify axonal degeneration, applying this methodology extensively to various neurological disorders. The robust adaptability of this experimental system is demonstrated by its application in exploring a wide spectrum of genetic mutations associated with neurological conditions, highlighting its broad utility in neurogenetic research. We identified a novel de novo variant in Spliceosome Associated Factor 1, Recruiter of U4/U6.U5 Tri-SnRNP (*SART1*). The patient, born at 37 weeks with a birth weight of 2934 g, exhibited significant developmental delays, including an inability to support head movement at 7 months, reliance on tube feeding, unresponsiveness to visual stimuli, and development of infantile spasms with hypsarrhythmia, as evidenced by EEG findings. Profound hearing loss and brain atrophy were confirmed through MRI imaging. To assess the functional impact of this novel human gene variant, we engineered transgenic *Drosophila* lines expressing both WT and mutant *SART1* under the control of a UAS promoter. Our MeDUsA analysis suggested that the variant may confer a gain-of-toxic-function (*Nitta et al., 2023*). Moreover, we identified heterozygous LOF mutations in *DHX9* as potentially causative for a newly characterized neurodevelopmental disorder. We further investigated the pathogenic potential of a novel heterozygous de novo missense mutation in *DHX9* in a patient presenting with short stature, intellectual disability, and myocardial compaction. Our findings indicated a LOF in the G414R and R1052Q variants of *DHX9* (*Yamada et al., 2023*). This experimental framework has been instrumental in elucidating the impact of gene mutations, enhancing our ability to diagnose how novel variants influence gene function. Our research established that *dOPA1* knockdown precipitates axonal degeneration and elevates ROS signals in retinal axons. Expression of human *OPA1* within this context effectively mitigated both phenomena; it partially reversed axonal degeneration and nearly completely normalized ROS levels. These results imply that factors other than increased ROS may drive the axonal degeneration observed post knockdown. Furthermore, while differences between the impacts of DN mutations and LOF mutations were evident in axonal degeneration, they were less apparent when using ROS as a biomarker. The extensive use of transgenes in our experiments might have mitigated the knockdown effects. In a systemic *dOPA1* knockdown, assessments of mitochondrial quantity and autophagy activity revealed no significant changes, suggesting that the cellular consequences of reduced *OPA1* expression might vary across different cell types.

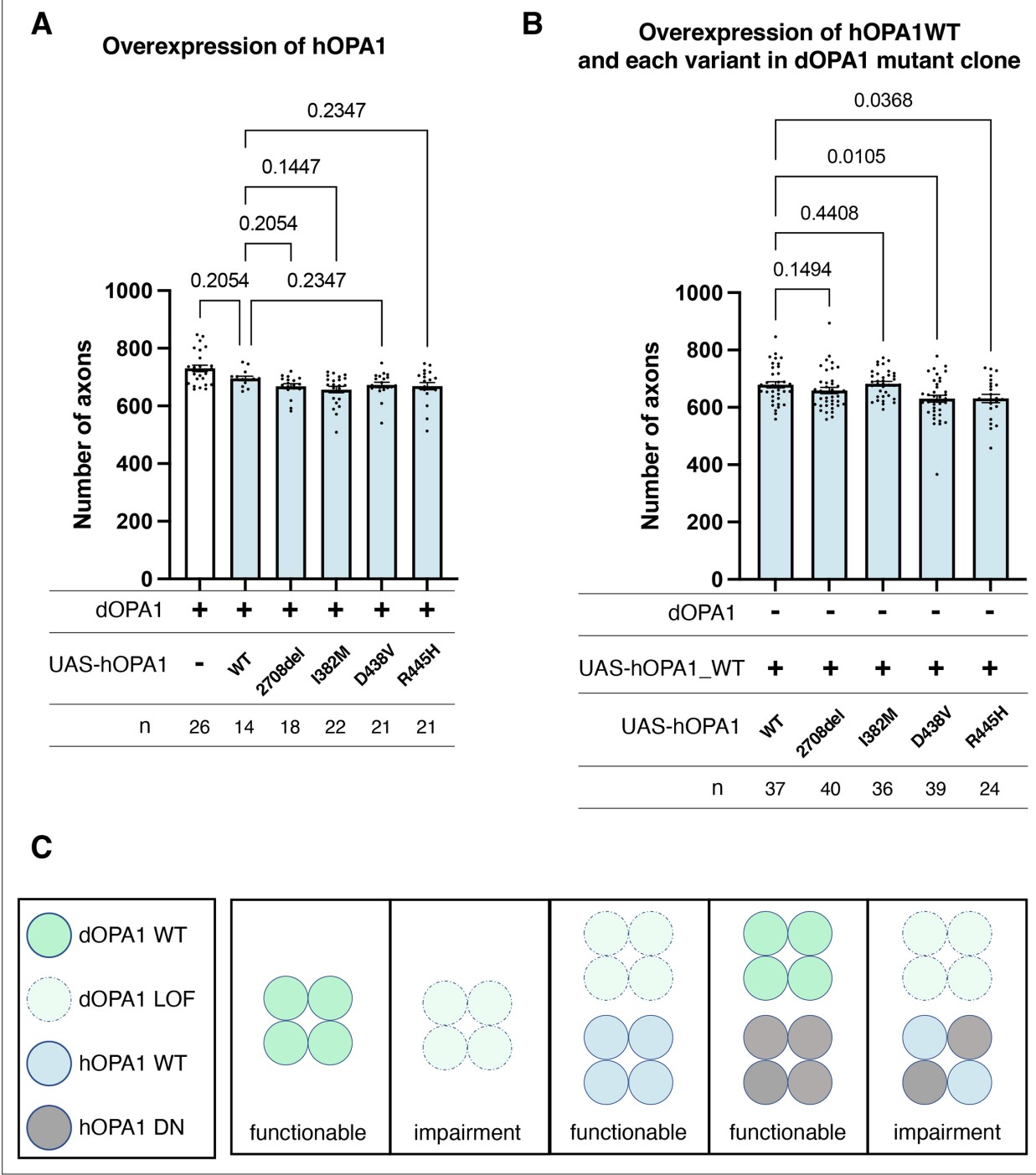

**Figure 6.** Loss-of-function (LOF) or dominant-negative (DN) effects of disease mutations in *hOPA1*. (**A**) Impact of each human OPA1 variant on the axon number in the optic nerve of *Drosophila* as quantified using MeDUsA (method for the quantification of degeneration using fly axons). (**B**) Expression of both *hOPA1* wild-type (WT) and its respective variants was analyzed in photoreceptors lacking *dOPA1*, and the number of axons was quantified using the MeDUsA. The sample size is indicated (n). Data are expressed as mean ± SEM. (**C**) Schematic representation of interspecies differences in OPA1 interactions and the interchangeability of OPA1 between human and fly. See the *Supplementary file 1* for the genotypes of the *Drosophila* used in the experiments.

The online version of this article includes the following figure supplement(s) for figure 6:

**Figure supplement 1.** Pairwise alignment of GTPase, Middle, and GED domains of hOPA1 and dOPA1.

**Figure supplement 2.** Comparative analysis of mitochondrial reactive oxygen species (ROS) levels, COXII expression, and autophagy between *hOPA1* wild-type and the variants expression under *dOPA1* knockdown conditions.

**Figure supplement 3.** The effect of a dominant-negative mutation in *dOPA1*.

In our study, expressing *dOPA1* in the retinal axons of *dOPA1* mutants resulted in significant rescue, but it did not return to control levels. There are three possible explanations for this result. The first concerns gene expression levels. The Gal4-line used for the rescue experiments may not replicate the expression levels or timing of endogenous *dOPA1*. Considering that the optimal functionality of *dOPA1* may be contingent upon specific gene expression levels, attaining a WT-like state necessitates the precise regulation of these expression levels. The second is a non-autonomous issue. Although *dOPA1* gene expression was induced in the retinal axons for the rescue experiments, many retinal axons were homozygous mutants, while other cell types were heterozygous for the *dOPA1* mutation. If there is a non-autonomous effect of *dOPA1* in cells other than retinal axons, it might not be possible to restore the WT-like state fully. The third potential issue is that only one isoform of *dOPA1* was expressed. In mouse OPA1, to completely restore mitochondrial network shape, an appropriate balance of at least two different isoforms, long-form OPA1 (L-OPA1) and short-form OPA1 (s-OPA1), is required (*Del Dotto et al., 2017*). This requirement implies that multiple isoforms of *dOPA1* are essential for the dynamic activities of mitochondria.

Our established fly model is the first simple organism to allow observation of degeneration of the retinal axons. The mitochondria in the axons showed fragmentation of mitochondria. Former studies have observed mitochondrial fragmentation in S2 cells (*McQuibban et al., 2006*), muscle tissue (*Deng et al., 2008*), segmental nerves (*Trevisan et al., 2018*), and ommatidia (*Yarosh et al., 2008*) due to the LOF of *dOPA1*. Our study presents compelling evidence that *dOPA1* knockdown instigates neuronal degeneration, characterized by a sequential deterioration at the axonal terminals and extending to the cell bodies. This degenerative pattern, commencing from the distal axons and progressing proximally toward the cell soma, aligns with the paradigm of 'dying-back' neuropathy, a phenomenon extensively documented in various neurodegenerative disorders (*Wang et al., 2012*). Previously, the function of *OPA1* in mice (*mOPA1*) was substituted by *hOPA1* (*Del Dotto et al., 2017*; *Sarzi et al., 2018*). Here, we demonstrated that the function of *dOPA1* can also be substituted by *hOPA1*. Thus, the function of *OPA1* is conserved across a wide range of species but our fly model has the advantage of allowing rapid observation of phenotypes. In the mouse model of DOA with the heterozygous deletion of the *mOPA1* gene at positions 329–355 (*Alavi et al., 2007*) or delTTAG mutant (*Sarzi et al., 2012*), a decrease in the number of RGC was observed after 17 months or 16 months of age, respectively. To save time, using a fly model for quick analysis and direct observation of retinal axons would be useful. Previous models using nematodes, flies, and zebrafish did not observe axonal degeneration directly, but our new model has made this possible. As a result, our model has the potential to be used for screening for modifiers of unclear molecular pathologies and drug screening.

Our model could also demonstrate the pathological significance of *hOPA1* mutations and be used for genetic diagnosis to differentiate between LOF and DN mutations. To elucidate the pathophysiological implications of mutations in the *OPA1* gene, we engineered and expressed several human *OPA1* variants, including the 2708-2711del mutation, associated with DOA, and the I382M mutation, located in the GTPase domain and linked to DOA. We also investigated the D438V and R445H mutations in the GTPase domain and correlated with the more severe DOA plus phenotype. The 2708-2711del mutation exhibited limited detectability via HA-tag probing. Still, it was undetectable with a myc tag, likely due to a frameshift event leading to the mutation's characteristic truncated protein product, as delineated in prior studies (*Zanna et al., 2008*). Contrastingly, the I382M, D438V, and R445H mutations demonstrated expression levels comparable to the WT *hOPA1*.

However, the expression of these mutants in retinal axons did not restore the *dOPA1* deficiency to the same extent as the WT hOPA1, as evidenced in *Figure 5E*. This finding indicates a functional impairment imparted by these mutations, aligning with established understanding (*Zanna et al., 2008*). Notably, while the 2708-2711del and I382M mutations exhibited limited functional rescue, the D438V and R445H mutations did not show significant rescue activity. This differential rescue efficiency suggests that the former mutations, particularly the I382M, categorized as a hypomorph (*Del Dotto et al., 2018*), may retain partial functional capacity, indicative of a LOF effect but with residual activity. The I382M missense mutation within the GTPase domain of OPA1 has been described as a mild hypomorph or a disease modifier. Intriguingly, this mutation alone does not induce significant clinical outcomes, as evidenced by multiple studies (*Bonifert et al., 2014*; *Bonneau et al., 2014*; *Carelli et al., 2015*; *Schaaf et al., 2011*). A significant reduction in protein levels has been observed

in fibroblasts originating from patients harboring the I382M mutation. However, mitochondrial volume remains unaffected, and the fusion activity of mitochondria is only minimally influenced (*Del Dotto et al., 2018*; *Kane et al., 2017*). This observation is consistent with findings reported by *Chao de la Barca et al., 2020*, where a targeted metabolomics approach classified I382M as a mild hypomorph. In our current study, the I382M mutation preserves more OPA1 function compared to DN mutations, as depicted in *Figure 5E and F*. Considering the results from our *Drosophila* model and previous research, we hypothesize that the I382M mutation may constitute a mild hypomorphic variant. This might explain its failure to manifest a phenotype on its own, yet its contribution to increased severity when it occurs in compound heterozygosity.

Our results for LOF or DN segmentation were consistent with previously reported yeast models (*Del Dotto et al., 2018*) and fibroblasts obtained from patients with OPA1 mutations (*Kane et al., 2017*), and confirmed the conserved function of *OPA1* across species. A major advantage of the *Drosophila* model over yeast or fibroblasts is that we can observe a phenotype that mimics the actual disease state of axonal degeneration in retinal axons and analyze it not only at the molecular level, but also the structural level of axons and mitochondria. The transgenic *Drosophila* of disease-associated *hOpa1* mutations created in this study can also serve to understand the pathophysiology of these mutations.

Advances in genome analysis allowed the identification of many variants and genetic mutations associated with DOA using next-generation sequencing. This could confirm diagnoses and identify new forms of DOA. The widespread use of these sequencers has also created new challenges, such as the growing number of variants of unknown significance (VUS) in publicly available databases such as CliniVar for DOA. These variants are categorized as pathogenic, likely pathogenic, VUS, likely benign, or benign, and the number of VUS is increasing as more variants are registered. The ClinVar database had classified 110 out of 319 *Opa1* variants as VUS at the end of 2019. By the end of 2022, this number had increased to 357 out of 840, indicating an increase from 34% to 42% over the last 3 years (*Henrie et al., 2018*). This trend suggests that the number of VUS in *Opa1* may continue to increase in the future. Analyzing VUS requires testing the effects of a mutation in vitro or in vivo; this can be expensive and time-consuming without prior confirmation or high probability of pathological significance. However, our fly model can be used to easily and inexpensively determine the pathological effects of disease-causing gene mutations among the growing number of rare mutations identified in DOA.

In this study, we tested the effect of expressing the D438V and R445H mutations in the *dOPA1* background and found that retinal axons did not degenerate. Regarding the interactions among OPA1 proteins, in yeast Mgm1, K854 in the region near the C-terminus is important for aggregation, and it becomes an oligomer to increase the activity of GTPase (*Rujiviphat et al., 2009*). Although hOPA1 normally has low GTPase activity, its interaction with negatively charged phospholipids such as cardiolipin causes hOPA1 to aggregate, increasing GTP hydrolysis activity (*Ban et al., 2010*). In the thermophilic fungus *Chaetomium thermophilum*, Mgm1 forms a dimer and then a tetramer at the C-terminal Stalk domain of Mgm1 (*Faelber et al., 2019*). Structural analysis also predicted that the Middle and GED domains are necessary for dimer and oligomer formation in hOPA1 (*Li et al., 2019*). In addition, peptide-binding assays have revealed that the coiled-coil domain, which is part of the GED domain, is required for hOPA1 interactions (*Akepati et al., 2008*). Comparing the amino acid sequences of hOPA1 and dOPA1, the Middle and GED domains, which are necessary for interactions between OPA1 molecules, exhibit concordance rates of 54.3% and 63.6%, respectively. These percentages are low compared to the 71.8% identity observed in the GTPase domains. These suggest that the OPA1 homologs of each species form oligomers with their unique OPA1. Note that the expression of $dOPA1^{K273A}$, a presumably GTPase-negative form of *dOPA1* (*Tsuyama et al., 2017*), degenerated retinal axons (*Figure 6—figure supplement 3*). These results imply that, while dOPA1 and hOPA1 are interchangeable, they may not interact with each other. This could pave the way for the development of a chimeric hOPA1, which would retain its functional properties while avoiding the interaction with endogenous hOPA1. Such chimeric hOPA1 could potentially be used as gene therapy since induced pluripotent stem cells have been generated from a patient with R445H (*Sladen et al., 2021*).

OPA1 has been implicated in various other diseases as well, including normal tension glaucoma (*Powell et al., 2003*; *Yu-Wai-Man et al., 2010b*), multiple sclerosis-like illness (*Verny et al., 2008*; *Yu-Wai-Man et al., 2016*), Parkinson's disease and dementia (*Carelli et al., 2015*; *Lynch et al., 2017*), and cardiomyopathy (*Spiegel et al., 2016*). Although the connection between these diseases and

OPA1 is not fully understood, further research on DOA may provide a deeper insight into this relationship. In the future, pathological analysis using the present model could have implications for understanding the mechanisms underlying these diseases.

DOA has also been reported to involve mtDNA depletion (*Amati-Bonneau et al., 2008*), but our model, while useful for nuclear gene analysis, is limited for mtDNA analysis. The gene content in the *Drosophila* mtDNA genome is similar as that in vertebrates, but the gene order and distribution on both DNA strands differ. mOPA1 interacts with mtDNA nucleoids through exon 4b binding to mtDNA D-loops, independent of mitochondrial fusion (*Yang et al., 2020*). However, the homologous region of mOPA1 or hOPA1 to exon 4b is not found in *Drosophila*, and the D-loop of mtDNA has not been found in cultured cells of flies or in mtDNA of embryos using electron microscopy (*Rubenstein et al., 1977*). In addition, while the D-loop is involved in mtDNA replication in humans (*Fish et al., 2004*), the A+T region is the origin of replication in mtDNA replication in *Drosophila* (*Goddard and Wolstenholme, 1978*). Thus, there may be differences in the mechanism of mtDNA replication between humans and flies (*Garesse and Kaguni, 2005*). Therefore, the association of dOPA1 with mtDNA may differ from that in mammals and the regulatory mechanism of mtDNA homeostasis in which dOPA1 is involved requires further investigation.

Another limitation of our model is that the molecular mechanisms underlying the formation of L-OPA1 and S-OPA1, which are well analyzed in mammals (*MacVicar and Langer, 2016*), have not been elucidated in *Drosophila*. In hOPA1, S1 or S2 sites are processed by the i-AAA protease OMA1 and Yme1L respectively to produce short form of hOPA1 (S-hOPA1) (*Anand et al., 2014*). Despite the fact that Yme1L is conserved in *Drosophila*, it has been involved in the degradation of the dOPA1 protein rather than its cleavage by dYme1L (*Liu et al., 2020*). Moreover, the cleavage site for i-AAA does not exist in dOPA1 (*Olichon et al., 2007*). Regarding other proteases, yeast Mgm1 is processed by the rhomboid protease Pcp1 (*Esser et al., 2002*; *Herlan et al., 2003*). However, the Rhomboid-7, a Pcp1 ortholog of *Drosophila*, was not required for dOPA1 processing (*Whitworth et al., 2008*). Although hOPA1 expression in *Drosophila* has identified three different sizes of hOPA1 in our western blotting result, it is unclear how they are cleaved. From a size perspective, the upper band was inferred to represent the full-length hOPA1 including the mitochondrial import sequence (MIS). Since the Middle and the lowest bands were not present in HA-tagged samples, it is possible that the bands at the Middle size represent the long-form hOPA1 in which probably MIS was processed. The band detected at the lowest position may represent an S-hOPA1. However, *Drosophila* does not have OMA1, an i-AAA protease that cleaves the S1 site. Although another i-AAA protease that cleaves the S2 site, YME1L, is conserved, our expressed hOPA1 is isoform 1 and lacks the S2 site (*Figure 5A*). Therefore, it is not clear how S-hOPA1 is processed in *Drosophila*. The anti-dOPA1 antiserum detects three bands in the overexpression of a FLAG-tagged dOPA1 construct (*Poole et al., 2010*), suggesting a molecular mechanism by which dOPA1, like yeast and mammals, is cleaved after translation. However, the proteases involved in the mechanism are not yet clarified. Whether *Drosophila's* endogenous mechanisms can be used to study how L-OPA1 and S-OPA1 are involved in DOA remains unclear.

In this study, we established a new model of DOA in *Drosophila*, in which we discovered the following: (1) We could replicate the human disease of optic nerve degeneration, allowing rapid genetic disease analysis. (2) We could distinguish the LOF and DN effects of *hOPA1* by observation of axonal degeneration phenotype but not with ROS signal, autophagy activity, and amount of the mitochondria in this model. These findings can reveal the pathological significance of de novo mutations and are useful in diagnosing DOA or DOA plus. Additionally, while *hOPA1* is the major gene involved in DOA, other genes involved in DOA and interacting proteins have not been investigated yet. Our model can be used for pathological modifier screening, exploring modifiers, and contributing to drug development. In the future, we hope to provide insights that can be applied to the fundamental treatment of DOA.

## Materials and methods

### Fly strains

Flies were maintained at 25°C on standard fly food. For knockdown experiments (*Figure 1C–E, F–J*, *Figure 2A–H*, *Figure 3B–K*, *Figure 5F*, *Figure 1—figure supplement 1*, *Figure 3—figure supplement 1*, *Figure 6—figure supplement 2A*, and *Figure 6—figure supplement 3*), flies were kept at

29°C in darkness. Female flies were used in all experiments to ensure consistency in the number of retinal axons. The following fly strains were obtained from the Bloomington *Drosophila* Stock Center (BDSC): *GMR-Gal4* (BDSC 1104), *Tub-Gal4* (BDSC 5138), *UAS-dicer2(III)* (BDSC 24651), *UAS-mito-HA-GFP* (BDSC 8443), *UAS-mCherry-mito* (BDSC 66532), *UAS-mitoQC* (BDSC 91641), *UAS-dOPA1 RNAi* (BDSC 32358), *UAS-Milton RNAi* (BDSC 44477), *FRT42D, GMR-hid, l(2)cl-R11(1)/CyO; ey-Gal4, UAS-flp* (BDSC 5251), *ey3.5-flp* (BDSC 35542), *Tub-Gal80^{TS}* (BDSC 7017), and *UAS-luciferase RNAi* (BDSC 31603). The following fly strains were obtained from the Vienna *Drosophila* Resource Center (VDRC): The *40D-UAS* (VDRC 60101) and *UAS-dOPA1 RNAi* (VDRC 106290). The following fly strains were obtained from the Kyoto *Drosophila* Stock Center (Kyoto): *FRT19* (Kyoto 101231), *FRT42D* (Kyoto 101878), and *dOPA1^{s3475}* (Kyoto 111438). The *UAS-dOPA1-HA* and *UAS-dOPA1-HA^{K273A}* were generously provided by Dr. T Uemura (Kyoto University, Japan). The *dOPA1* clone analysis was performed by inducing flippase expression in the eyes using either *ey-Gal4* with *UAS-flp* or *ey3.5-flp*, followed by recombination at the chromosomal location *FRT42D* to generate a mosaic of cells homozygous for *dOPA1^{s3475}*.

## Generation of transgenic flies

The cDNA for WT, I382M, D438V, R445H, and 2708-2711del forms of *hOPA1* (NM_015560.3) were generated by Vectorbuilder Inc (Yokohama, Japan). Then, the HA-hOPA1-myc sequence was amplified and inserted into pJFRC81-10XUAS-IVS-Syn21-GFP-p10 (ID36432, Addgene, USA) digested with *NotI* and *XbaI* using primers with the N-terminal containing the Kozak and HA sequences (TTACTTCA GGCGGCCGCGGCCAAAATGTACCCATACGATGTTCCAGATTAC) and the C-terminal containing the myc sequence (TTAAAAACGATTCATTCTAGTTACAGATCCTCTTCTGAGATGAGTTTTTGTTCTTT CTCCTGATGAAGAGCTTCAATG). To confirm production of the complete coding sequence, it was bidirectionally sequenced using the Sanger method. These plasmids were injected into embryos and integrated into the attP2 landing site via ΦC31 recombination (WellGenetics, Taiwan).

## Immunohistochemistry and imaging

Immunohistochemistry was performed as described previously (*Sugie et al., 2017*). Flies were anesthetized on a $CO_2$ pad, and specimens of the correct genotype were selected. Specimens were briefly washed in ethanol for a few seconds before being transferred to a Petri dish containing phosphate-buffered saline (PBS). Using forceps, the proboscis was secured, and the right and left eyes were dissected out with another set of forceps. The proboscis and attached trachea were also removed from the brain. The brains were then carefully transferred with forceps into a 1.5 mL tube containing 150 µL of 0.3% PBT (PBS with Triton X-100) at room temperature.

To fix the tissues, 50 µL of 16% formaldehyde was added to each tube, followed by gentle mixing. The samples were incubated in the fixative for 50 min at room temperature. Subsequent washing involved three rinses with 200 µL of 0.3% PBT, using a micropipette for fluid exchange. After the final wash, the PBT was removed, and the samples were incubated in the primary antibody solution overnight (O/N) at 4°C. This was followed by three washes with 0.3% PBT, after which the secondary antibody solution (diluted in 0.3% PBT) was added, and samples were incubated O/N in the dark at 4°C. After another three washes in 0.3% PBT, the samples were prepared for imaging. The following antibodies were used: rat anti-HA (3F10, 1:400; Roche, Switzerland), mouse anti-Chaoptin (24B10, 1:25; Developmental Studies Hybridoma Bank, USA), rat anti-CadN (DN-Ex#8, 1:50; Developmental Studies Hybridoma Bank), anti-rat Alexa Fluor 488 (1:400; Thermo Fisher Scientific, USA), anti-mouse Alexa Fluor 488 (1:400; Thermo Fisher Scientific), anti-mouse Alexa Fluor 568 (1:400; Thermo Fisher Scientific), and anti-rat Alexa Fluor 633 (1:400; Thermo Fisher Scientific). Specimens were mounted using a Vectashield mounting medium (Vector Laboratories, USA). Images were captured using an FV3000 confocal microscope (Olympus, Japan) or a C2 (Nikon, Japan) confocal microscope for *Figure 1—figure supplement 1* and processed using the IMARIS software (Oxford Instruments, UK) or Fiji software, an open-source image analysis software (*Schindelin et al., 2012*).

## Electron microscopy

Fly heads were fixed O/N with 2.5% glutaraldehyde and 2% paraformaldehyde in 0.1 M sodium cacodylate buffer. Heads were then post-fixed with 1% osmium tetroxide, dehydrated in ethanol, and embedded in Epon. Ultra-thin tangential sections of the laminas (70 nm) were stained with uranyl

acetate and lead citrate. Random retinas or lamina sections were imaged at a magnification of ×100k with a VELETA CCD Camera (Olympus) mounted on a JEM 1010 transmission electron microscope (JEOL, Japan). In the R1–6 axons of the lamina, the areas corresponding to mitochondria were quantified using the freehand line tool in NIS Elements software (Nikon). Concurrently, mitochondria were manually categorized based on their structural characteristics: those with a densely packed matrix were classified as Class A, and those with a collapsed structure as Class B.

## Quantification of Mito-GFP intensity in retinal axons and retina

To measure the relative intensity of Mito-GFP in the photoreceptor R7/8 axons of the medulla, the imaging analysis software IMARIS was used. First, the M1–6 layers of the medulla, including the R7/8 axons, were manually covered using the surface function. Then, the surface area was masked, the surface of the left axons was stained with anti-Chaoptin, and the Mito-GFP signal was generated automatically using the surface function. Finally, the voxel number of the Mito-GFP surface was divided by the voxel number of the R axon surface. These quantifications were performed by experimenters blind to the genotype.

## Mitochondrial superoxide detection and quantification

The ROS level was determined using the MitoSOX Red Mitochondrial Superoxide Indicator (Thermo Fisher Scientific). The procedure involved dissecting adult brains in PBS, transferring them to 5 µM MitoSOX solution, and incubating them for 10 min in the dark at room temperature. The brains were then washed with PBS containing 20% Vectashield (Vector Laboratories), followed by sequential washes with PBS containing 40% and 60% Vectashield. The samples were mounted in Vectashield and scanned using an FV3000 confocal microscope (Olympus). To quantify the ROS level in photoreceptor R7/8 axons, the IMARIS software (Bitplane) was used for imaging analysis. The software generated automatic surfaces of the MitoSOX and Mito-GFP signals. Then, the ROS level was calculated by dividing the voxel number of the MitoSOX surface on the Mito-GFP surface by the voxel number of the Mito-GFP surface. These quantifications were performed by experimenters blind to the genotype.

## Measurement of mitophagy levels in retinal axons

The mitophagy levels in the photoreceptor R7/8 axons of the medulla were analyzed using the mitoQC reporter. Mitolysosomes were identified by referencing the red-only mCherry puncta of mitoQC. The number of mitolysomes of each voxel image was manually quantified with IMARIS software (Bitplane). These quantifications were performed by experimenters blind to the genotype.

## Western blot analysis of the whole *Drosophila* body

We regulated protein expression temporally across the whole body using the *Tub-Gal4* and *Tub-GAL80^TS* system. Flies harboring each *hOPA1* variant were maintained at a permissive temperature of 20°C, and upon emergence, females were transferred to a restrictive temperature of 29°C for subsequent experiments. The flies were collected in plastic tubes and sonicated (2×30 s) using a Q55 sonicator (Qsonica, Newtown, CT, USA) in lysis buffer (10 mM Tris–HCl, pH 7.5, 150 mM NaCl, 1 mM EDTA, 2% DDM: 10 µL/mg flies) supplemented with 1:1000 (vol/vol) Protease Inhibitor Cocktail Set III (Calbiochem, USA). Rat anti-HRP-conjugated-HA (3F10, 1: 3000; Proteintech, Germany), mouse anti-myc (9B11, 1: 10000; Cell Signaling Technology, USA), rabbit anti-OPA1 (ab42364, 1: 1000; Abcam, UK), rabbit anti-COXII (1: 100; *Murari et al., 2020*), rabbit anti-Atg8a (ab109364, 1: 8000; Abcam), mouse anti-α-Tubulin (T9026, 1:100,000; Sigma-Aldrich, USA), goat HRP-conjugated anti-mouse IgG (SA00001-1, 1: 10,000; Proteintech), and goat HRP-conjugated anti-rabbit IgG (7074S, 1: 5,000; Cell Signaling Technology) antibodies were used for western blotting.

## Pairwise alignment of hOPA1 and dOPA1

The amino acid sequences for hOPA1 and dOPA1 were referenced from NM_015560.3 and NM_166040.2, respectively, and aligned using the EMBOSS Needle (*Rice et al., 2000*). The GTPase, Middle, and GED domains in dOPA1 were identified by cross-referencing the previously described amino acid positions of these domains in hOPA1 (*Liesa et al., 2009*). The degree of identity between the aligned amino acid sequences was subsequently determined.

## Experimental design and statistical analyses

Experimental analyses were performed using Prism 9 (GraphPad Software, USA). For *Figures 1E, H and 2C*, *Figure 3—figure supplement 1B*, and *Figure 6—figure supplement 3B*, data were analyzed using t-tests (and nonparametric tests) and Mann-Whitney tests. For *Figure 1J*, Fisher's exact test was used. For *Figures 2H, 3F, K–5C, 5E, F, 6A and B*, *Figure 3—figure supplement 2B*, *Figure 5—figure supplement 1B*, *Figure 5—figure supplement 1D*, *Figure 6—figure supplement 2C*, and *Figure 6—figure supplement 2E*, differences among multiple groups were examined using the Kruskal–Wallis test followed by the two-stage linear step-up procedure of Benjamini, Krieger, and Yekutieli tests. The null hypothesis was rejected at a 0.05 level of significance. Sample sizes are indicated in each figure. The number of retinal axons and rhabdomeres, the intensity of Mito-GFP and MitoSOX, the mitochondrial area and the classification, and the corrected p-values are described in the Results section. All data are expressed as the mean ± SEM.

## Acknowledgements

We would like to thank Dr. Uemura for providing us with the transgenic fly strain and Dr. Owusu-Ansah for gifting us the antibody. The Ministry of Education, Culture, Sports, Science, and Technology of Japan (#18K14835, #18J00367, and #21K15619 to YN, #21K06184 to SHS, #16H06457, #21H02483, and 21H05682 to TS, #17H04983, #19K22592, and #21H02837 to AS), the Japan Agency for Medical Research and Development (AMED) (#JP22ek019484s) to AS, NIG-JOINT to ES, Takeda Science Foundation Takeda Visionary Research Grant to TS, Takeda Science Foundation Life Science Research Grant to AS, and GSK Japan Research Grant 2022 (AS2021A000166849) to AS.

## Additional information

### Funding

| Funder | Grant reference number | Author |
|---|---|---|
| Japan Society for the Promotion of Science | 18K14835 | Yohei Nitta |
| Japan Society for the Promotion of Science | 18J00367 | Yohei Nitta |
| Japan Society for the Promotion of Science | 21K15619 | Yohei Nitta |
| Japan Society for the Promotion of Science | 21K06184 | Satoko Hakeda-Suzuki |
| Japan Society for the Promotion of Science | 16H06457 | Takashi Suzuki |
| Japan Society for the Promotion of Science | 21H02483 | Takashi Suzuki |
| Japan Society for the Promotion of Science | 21H05682 | Takashi Suzuki |
| Japan Society for the Promotion of Science | 17H04983 | Atsushi Sugie |
| Japan Society for the Promotion of Science | 19K22592 | Atsushi Sugie |
| Japan Society for the Promotion of Science | 21H02837 | Atsushi Sugie |
| Japan Society for the Promotion of Science | 24K02349 | Atsushi Sugie |
| Japan Society for the Promotion of Science | 24K22104 | Atsushi Sugie |

| Funder | Grant reference number | Author |
| --- | --- | --- |
| Japan Agency for Medical Research and Development | JP22ek019484s | Atsushi Sugie |
| Japan Agency for Medical Research and Development | JP24ek0109760s4001 | Atsushi Sugie |
| NIG-JOINT | | Emiko Suzuki |
| Takeda Science Foundation | Visionary Research Grant | Takashi Suzuki |
| Takeda Science Foundation | Life Science Research Grant | Atsushi Sugie |
| Takeda Science Foundation | Bioscience Research Grant | Atsushi Sugie |
| GSK Japan Research Grant 2022 | AS2021A000166849 | Atsushi Sugie |

The funders had no role in study design, data collection and interpretation, or the decision to submit the work for publication.

## Author contributions

Yohei Nitta, Conceptualization, Data curation, Formal analysis, Funding acquisition, Validation, Investigation, Visualization, Methodology, Writing - original draft, Writing - review and editing; Jiro Osaka, Data curation, Formal analysis, Validation, Investigation, Visualization, Methodology, Writing - original draft, Writing - review and editing; Ryuto Maki, Formal analysis, Investigation, Methodology; Satoko Hakeda-Suzuki, Formal analysis, Supervision, Funding acquisition, Investigation, Methodology; Emiko Suzuki, Takashi Suzuki, Supervision, Funding acquisition; Satoshi Ueki, Supervision; Atsushi Sugie, Conceptualization, Data curation, Formal analysis, Supervision, Funding acquisition, Validation, Investigation, Visualization, Methodology, Writing - original draft, Project administration, Writing - review and editing

## Author ORCIDs

Yohei Nitta https://orcid.org/0000-0002-0712-428X
Jiro Osaka http://orcid.org/0009-0008-8015-1955
Emiko Suzuki https://orcid.org/0000-0002-4005-0542
Takashi Suzuki https://orcid.org/0000-0001-9093-2562
Atsushi Sugie https://orcid.org/0000-0002-2090-8839

Reviewer #3 (Public Review): https://doi.org/10.7554/eLife.87880.3.sa1
Author response https://doi.org/10.7554/eLife.87880.3.sa2

# Additional files

## Supplementary files

• Supplementary file 1. The genotypes of the *Drosophila* used in the experiments. Detailed information about the specific genotypes of *Drosophila* used in each experiment described in the manuscript, including the relevant genetic constructs, mutations, and background strains.

• MDAR checklist

## Data availability

The computer code essential for replicating the axonal degeneration quantification of this study is available on GitHub (copy archived at *Kawai et al., 2024*). This study does not involve any specific datasets. All relevant data generated or analyzed during this study are included in the manuscript itself. No additional data files are applicable for this research.

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
