## [Editor Report · eLife assessment]

This study provides **valuable** insights into the complex genetics of dominant optic atrophy. Leveraging a fly model, the investigators provide **solid** evidence, albeit with small effect sizes, for a dominant negative mechanism of certain pathogenic variants that tend to cause more severe phenotypes, a long held hypothesis in the field. The work is of high interest to those in the optic atrophy and degeneration fields.

---

## [Referee Report · Reviewer #3 (Public Review)]

Nitta et al. use a fly model of autosomal dominant optic atrophy to provide mechanistic insights into distinct disease-causing OPA1 variants. It has long been hypothesized that missense OPA1 mutations affecting the GTPase domain, which are associated with more severe optic atrophy and extra-ophthalmic neurologic conditions such as sensorineural hearing loss (DOA plus), impart their effects through a dominant negative mechanism, but no clear direct evidence for this exists particularly in an animal model. The authors execute a well-designed study to establish their model, demonstrating a mitochondrial phenotype and optic atrophy measured as axonal degeneration. They leverage this model to provide the first direct evidence for a dominant negative mechanism for 2 mutations causing DOA plus by expressing these variants in the background of a full hOPA1 complement.

Strengths of the paper include well-motivated objectives and hypotheses, and overall solid design and execution. There is a thorough discussion of the interpretation and context of the findings. The results technically support their primary conclusions with minor limitations. First, while only partial rescue of the most clinically relevant metric for optic atrophy in this model is now acknowledged, the result nevertheless hamstrings the mechanistic experiments that follow. Second, the results statistically support a dominant negative effect of DOA plus-associated variants, yet the data show a marginal impact on axonal degeneration for these variants. In added experiments, the ability of WT hOPA1 and I382M but not 2708del, D438V or R445H to rescue ROS levels or mitophagy in the context of dOPA1 knockdown serves to support axonal number as a valid measure of mitochondrial function in this context. However, the critical experiment demonstrating a dominant negative effect was performed in the context of expressing WT hOPA1 along with a pathogenic variant, in which no differences in ROS, COXII expression or mitophagy were seen. This makes it difficult to conclude that the dominant negative effect of D438V and R445H on axon number is related to mitochondrial function.

As an animal model of DOA that may serve for rapid assessment of suspected OPA1 variants, the results overall support utility of this model in identifying pathogenic variants but not in distinguishing haploinsufficiency from dominant negative mechanisms among those variants. The impact of this work in providing the first direct evidence of a dominant negative mechanism is under-stated considering how important this question is in development of genetic treatments for dominant optic atrophy.

**Comments on revised version:**

The authors have addressed the comments in my initial review. Through these modification and those related to the comments from the other reviewers, the manuscript is strengthened.

**Comments on author responses to each of the reviews:**

Reviewer 1:

Interpretation of data has been appropriately reorganized in the discussion.

Quantified mitochondria in the model show no difference in number. There is reduced size and structural abnormalities on electron microscopy.

Application of mito-QC revealed increased mitophagy.

Regarding partial rescue of axonal number in the mutant model, statistical significance between control and rescue is still not depicted in Figure 4D. Detailing possible explanations for this has been addressed in the discussion. However, only partial rescue of the most clinically relevant metric for optic atrophy in this model hamstrings subsequent mechanistic experiments that follow.

Discussion regarding variant I382M has been improved.

While reviewer 1's concerns about axonal number as a biomarker for OPA1 function are valid, it is worth noting that this is the most clinically relevant marker in the context of DOA. That said, I agree that the mechanistic DN/HI studies needed support using other measures of mitochondrial function, and the authors have done this. The ability of WT hOPA1 and I382M but not 2708del, D438V or R445H to rescue ROS levels or mitophagy in the context of dOPA1 knockdown serves to support axonal number as a valid measure of mitochondrial function in this context. However, the critical experiment demonstrating a dominant negative effect was performed in the context of expressing WT hOPA1 along with a pathogenic variant, in which no differences in ROS, COXII expression or mitophagy were seen. This makes it difficult to conclude that the (marginal) DN effect of D438V and R445H on axon number is related to mitochondrial function, and serves as a minor weakness of the paper.

Which exons are included in the transcript, and therefore, which isoforms are expressed in the model, has been addressed.

Reviewer 2:

The authors have addressed the need to include greater methodological details.

Language concerning the clinical utility of the model in informing treatment decisions has been appropriately modified. As pointed out by Reviewer 1, additional studies were needed to better establish the potential clinical utility of this model in screening DOA variants. The authors have completed those experiments, and the results overall support utility of this model in identifying pathogenic variants but not in distinguishing HI/DN mechanisms among those variants.

Reviewer 3:

The author has addressed the partial rescue effect as above.

The authors have not modified the text to acknowledge the marginal effect sizes in the critical experiment of the study that demonstrates a DN effect. Statistically, the results indeed support a dominant negative effect of DOA plus-associated variants, yet the data show a marginal impact on axonal degeneration for these variants. This remains a weakness of the study.

---

## [Author Response]

The following is the response to the original reviews.

**Public Reviews:**

**Reviewer #1 (Public Review):**
Nitta et al, in their manuscript titled, "*Drosophila* model to clarify the pathological significance of OPA1 in autosomal dominant optic atrophy." The novelty of this paper lies in its use of human (hOPA1) to try to rescue the phenotype of an OPA1 +/- *Drosophilia* DOA model (dOPA). The authors then use this model to investigate the differences between dominant-negative and haploinsufficient OPA1 variants. The value of this paper lies in the study of DN/HI variants rather than the establishment of the *Drosophila* model per se as this has existed for some time and does have some significant disadvantages compared to existing models, particularly in the extra-ocular phenotype which is common with some OPA1 variants but not in humans. I judge the findings of this paper to be valuable with regards to significance and solid with regards to the strength of the evidence.Suggestions for improvements:(1) Stylistically the results section appears to have significant discussion/conclusion/inferences in section with reference to existing literature. I feel that this information would be better placed in the separate discussion section. E.g. lines 149-154.

We appreciate the reviewer’s suggestion to relocate the discussion, conclusions, and inferences, particularly those that reference existing literature, to a separate discussion section. For lines 149–154, we placed them in the discussion section (lines 343–347) as follows. “Our established fly model is the first simple organism to allow observation of degeneration of the retinal axons. The mitochondria in the axons showed fragmentation of mitochondria. Former studies have observed mitochondrial fragmentation in S2 cells (McQuibban et al., 2006), muscle tissue (Deng et al., 2008), segmental nerves (Trevisan et al., 2018), and ommatidia (Yarosh et al., 2008) due to the LOF of *dOPA1*.”

For lines 178–181, we also placed them in the discussion section (lines 347–351) as follows. “Our study presents compelling evidence that *dOPA1* knockdown instigates neuronal degeneration, characterized by a sequential deterioration at the axonal terminals and extending to the cell bodies. This degenerative pattern, commencing from the distal axons and progressing proximally towards the cell soma, aligns with the paradigm of 'dying-back' neuropathy, a phenomenon extensively documented in various neurodegenerative disorders (Wang et al., 2012). ”

For lines 213–217, 218–220, and 222–223, we also placed them in the discussion section (lines 363– 391) as follows. “To elucidate the pathophysiological implications of mutations in the *OPA1* gene, we engineered and expressed several human *OPA1* variants, including the 2708-2711del mutation, associated with DOA, and the I382M mutation, located in the GTPase domain and linked to DOA. We also investigated the D438V and R445H mutations in the GTPase domain and correlated with the more severe DOA plus phenotype. The 2708-2711del mutation exhibited limited detectability via HA-tag probing. Still, it was undetectable with a myc tag, likely due to a frameshift event leading to the mutation's characteristic truncated protein product, as delineated in prior studies (Zanna et al., 2008). Contrastingly, the I382M, D438V, and R445H mutations demonstrated expression levels comparable to the WT *hOPA1*. However, the expression of these mutants in retinal axons did not restore the dOPA1 deficiency to the same extent as the WT hOPA1, as evidenced in Figure 5E. This finding indicates a functional impairment imparted by these mutations, aligning with established understanding (Zanna et al., 2008). Notably, while the 2708-2711del and I382M mutations exhibited limited functional rescue, the D438V and R445H mutations did not show significant rescue activity. This differential rescue efficiency suggests that the former mutations, particularly the I382M, categorized as a hypomorph (Del Dotto et al., 2018), may retain partial functional capacity, indicative of a LOF effect but with residual activity. The I382M missense mutation within the GTPase domain of OPA1 has been described as a mild hypomorph or a disease modifier. Intriguingly, this mutation alone does not induce significant clinical outcomes, as evidenced by multiple studies (Schaaf et al., 2011; Bonneau et al., 2014; Bonifert et al., 2014; Carelli et al., 2015). A significant reduction in protein levels has been observed in fibroblasts originating from patients harboring the I382M mutation. However, mitochondrial volume remains unaffected, and the fusion activity of mitochondria is only minimally influenced (Kane et al., 2017; Del Dotto et al., 2018). This observation is consistent with findings reported by de la Barca et al. in Human Molecular Genetics 2020, where a targeted metabolomics approach classified I382M as a mild hypomorph. In our current study, the I382M mutation preserves more OPA1 function compared to DN mutations, as depicted in Figures 5E and F. Considering the results from our *Drosophila* model and previous research, we hypothesize that the I382M mutation may constitute a mild hypomorphic variant. This might explain its failure to manifest a phenotype on its own, yet its contribution to increased severity when it occurs in compound heterozygosity.

(2) I do think further investigation as to why a reduction of mitochondria was noticed in the knockdown. There are conflicting reports on this in the literature. My own experience of this is fairly uniform mitochondrial number in WT vs OPA1 variant lines but with an increased level of mitophagy presumably reflecting a greater turnover. There are a number of ways to quantify mitochondrial load e.g. mtDNA quantification, protein quantification for tom20/hsp60 or equivalent. I feel the reliance on ICC here is not enough to draw conclusions. Furthermore, mitophagy markers could be checked at the same time either at the transcript or protein level. I feel this is important as it helps validate the *Drosophila* model as we already have a lot of experimental data about the number and function of mitochondria in OPA+/- human/mammalian cells.

We thank the reviewer for the insightful comments and suggestions regarding our study on the impact of mitochondrial reduction in a knockdown model. We concur with the reviewer’s observation that our initial results did not definitively demonstrate a decrease in the number of mitochondria in retinal axons. Furthermore, we measured mitochondrial quantity by conducting western blotting using antiCOXII and found no reduction in mitochondrial content with the knockdown of *dOPA1* (Figure S4A and B). Consequently, we have revised our manuscript to remove the statement “suggesting a decreased number of mitochondria in retinal axons. However, whether this decrease is due to degradation resulting from a decline in mitochondrial quality or axonal transport failure remains unclear.” Instead, we have refocused our conclusion to reflect our electron microscopy findings, which indicate reduced mitochondrial size and structural abnormalities. The reviewer’s observation of consistent mitochondrial numbers in WT versus mutant variant lines and elevated mitophagy levels prompted us to evaluate mitochondrial turnover as a significant factor in our study. Regarding verifying mitophagy markers, we incorporated the mito-QC marker in our experimental design. In our experiments, mito-QC was expressed in the retinal axons of *Drosophila* to assess mitophagy activity upon *dOPA1* knockdown. We observed a notable increase in mCherry positive but GFP negative puncta signals one week after eclosion, indicating the activation of mitophagy (Figure 2D–H). This outcome strongly suggests that *dOPA1* knockdown enhances mitophagy in our *Drosophila* model. The application of mito-QC as a quantitative marker for mitophagy, validated in previous studies, offers a robust approach to analyzing this process. Our findings elucidate the role of *dOPA1* in mitochondrial dynamics and its implications for neuronal health. These results have been incorporated into Figure 2, with the corresponding text updated as follows (lines 159–167): “Given that an increase in mitophagy activity has been reported in mouse RGCs and nematode ADOA models (Zaninello et al., 2022; Zaninello et al., 2020), the mitoQC marker, an established indicator of mitophagy activity, was expressed in the photoreceptors of *Drosophila*. The mito-QC reporter consists of a tandem mCherry-GFP tag that localizes to the outer membrane of mitochondria (Lee et al., 2018). This construct allows the measurement of mitophagy by detecting an increase in the red-only mCherry signal when the GFP is degraded after mitochondria are transported to lysosomes. Post *dOPA1* knockdown, we observed a significant elevation in mCherry positive and GFP negative puncta signals at one week, demonstrating an activation of mitophagy as a consequence of *dOPA1* knockdown (Figure 2D–H).”

(3) Could the authors comment on the failure of the dOPA1 rescue to return their biomarker, axonal number to control levels. In Figure 4D is there significance between the control and rescue. Presumably so as there is between the mutant and rescue and the difference looks less.

As the reviewer correctly pointed out, there is a significant difference between the control and rescue groups, which we have now included in the figure. Additionally, we have incorporated the following comments in the discussion section (lines 329–342) regarding this significant difference: “In our study, expressing *dOPA1* in the retinal axons of *dOPA1* mutants resulted in significant rescue, but it did not return to control levels. There are three possible explanations for this result. The first concerns gene expression levels. The Gal4-line used for the rescue experiments may not replicate the expression levels or timing of endogenous *dOPA1*. Considering that the optimal functionality of *dOPA1* may be contingent upon specific gene expression levels, attaining a wild-type-like state necessitates the precise regulation of these expression levels. The second is a nonautonomous issue. Although *dOPA1* gene expression was induced in the retinal axons for the rescue experiments, many retinal axons were homozygous mutants, while other cell types were heterozygous for the *dOPA1* mutation. If there is a non-autonomous effect of dOPA1 in cells other than retinal axons, it might not be possible to restore the wild-type-like state fully. The third potential issue is that only one isoform of *dOPA1* was expressed. In mouse OPA1, to completely restore mitochondrial network shape, an appropriate balance of at least two different isoforms, lOPA1 and s-OPA1, is required (Del Dotto et al., 2017). This requirement implies that multiple isoforms of *dOPA1* are essential for the dynamic activities of mitochondria.”

(4) The authors have chosen an interesting if complicated missense variant to study, namely the I382M with several studies showing this is insufficient to cause disease in isolation and appears in high frequency on gnomAD but appears to worsen the phenotype when it appears as a compound het. I think this is worth discussing in the context of the results, particularly with regard to the ability for this variant to partially rescue the dOPA1 model as shown in Figure 5.

As the reviewer pointed out, the I382M mutation is known to act as a disease modifier. However, in our system, as suggested by Figure 5, I382M appears to retain more activity than DN mutations. Considering previous studies, we propose that I382M represents a mild hypomorph. Consequently, while I382M alone may not exhibit a phenotype, it could exacerbate severity in a compound heterozygous state. We have incorporated this perspective in our revised discussion (lines 375-391).

“Notably, while the 2708-2711del and I382M mutations exhibited limited functional rescue, the D438V and R445H mutations did not show significant rescue activity. This differential rescue efficiency suggests that the former mutations, particularly the I382M, categorized as a hypomorph (Del Dotto et al., 2018), may retain partial functional capacity, indicative of a LOF effect but with residual activity. The I382M missense mutation within the GTPase domain of OPA1 has been described as a mild hypomorph or a disease modifier. Intriguingly, this mutation alone does no induce significant clinical outcomes, as evidenced by multiple studies (Schaaf et al., 2011; Bonneau et al., 2014; Bonifert et al., 2014; Carelli et al., 2015). A significant reduction in protein levels has been observed in fibroblasts originating from patients harboring the I382M mutation. However, mitochondrial volume remains unaffected, and the fusion activity of mitochondria is only minimally influenced (Kane et al., 2017; Del Dotto et al., 2018). This observation is consistent with findings reported by de la Barca et al. in Human Molecular Genetics 2020, where a targeted metabolomics approach classified I382M as a mild hypomorph. In our current study, the I382M mutation preserves more OPA1 function compared to DN mutations, as depicted in Figures 5E and F. Considering the results from our *Drosophila* model and previous research, we hypothesize that the I382M mutation may constitute a mild hypomorphic variant. This might explain its failure to manifest a phenotype on its own, yet its contribution to increased severity when it occurs in compound heterozygosity.”

(5) I feel the main limitation of this paper is the reliance on axonal number as a biomarker for OPA1 function and ultimately rescue. I have concerns because (a) this is not a well validated biomarker within the context of OPA1 variants (b) we have little understanding of how this is affected by over/under expression and (c) if it is a threshold effect e.g. once OPA1 levels reach <x% pathology develops but develops normally when opa1 expression is >x%. I think this is particularly relevant when the authors are using this model to make conclusions on dominant negativity/HI with the authors proposing that if expression of a hOPA1 transcript does not increase opa1 expression in a dOPA1 KO then this means that the variant is DN. The authors have used other biomarkers in parts of this manuscript e.g. ROS measurement and mito trafficking but I feel this would benefit from something else particularly in the latter experiments demonstrated in figure 5 and 6.

The reviewer raised concerns regarding the adequacy of axonal count as a validated biomarker in the context of *OPA1* mutants. In response, we corroborated its validity using markers such as MitoSOX, Atg8, and COXII. Experiments employing MitoSOX revealed that the augmented ROS signals resulting from *dOPA1* knockdown were mitigated by expressing human *OPA1*. Conversely, the mutant variants 2708-2711del, D438V, and R445H did not ameliorate these effects, paralleling the phenotype of axonal degeneration observed. These findings are documented in Figure 5F, and we have incorporated the following text into section lines 248–254 of the results:

“Furthermore, we assessed the potential for rescuing ROS signals. Similar to its effect on axonal degeneration, wild-type hOPA1 effectively mitigated the phenotype, whereas the 2708-2711del, D438V, and R445H mutants did not (Figure 5F). Importantly, the I382M variant also reduced ROS levels comparably to the wild type. These findings demonstrate that both axonal degeneration and the increase in ROS caused by dOPA1 downregulation can be effectively counteracted by hOPA1. Although I382M retains partial functionality, it acts as a relatively weak hypomorph in this experimental setup.”

Moreover, utilizing mito-QC, we observed elevated mitophagy in our *Drosophila* model, with these results now included in Figure 2D–H. Given the complexity of the genetics involved and the challenges in establishing lines, autophagy activity was quantified by comparing the ratio of Atg8-1 to Atg8-2 via Western blot analysis. However, no significant alterations were detected across any of the genotypes. Additionally, mitochondrial protein levels derived from COXII confirmed consistent mitochondrial quantities, showing no considerable variance following knockdown. These insights affirm that retinal axon degeneration and mitophagy activation are present in the *Drosophila* DOA model, although the Western blot analysis revealed no significant changes in autophagy activation. Such findings necessitate caution as this model may not fully replicate the molecular pathology of the corresponding human disease. These Western blot findings are presented in Figure S4, with the following addition made to section lines 255–263 of the results:

“We also conducted Western blot analyses using anti-COXII and anti-Atg8a antibodies to assess changes in mitochondrial quantity and autophagy activity following the knockdown of *dOPA1*. Mitochondrial protein levels, indicated by COXII quantification, were evaluated to verify mitochondrial content, and the ratio of Atg8a-1 to Atg8a-2 was used to measure autophagy activation. For these experiments, *Tub-Gal4* was employed to systemically knockdown *dOPA1.* Considering the lethality of a whole-body *dOPA1* knockdown, *Tub-Gal80TS* was utilized to repress the knockdown until eclosion by maintaining the flies at 20°C. After eclosion, we increased the temperature to 29°C for two weeks to induce the knockdown or expression of *hOPA1* variants. The results revealed no significant differences across the genotypes tested (Figure S4A–D).”

In assessing the effects of dominant negative mutations, measurements including ROS levels, the ratio of Atg8-1 to Atg8-2, and the quantity of COXII protein were conducted, yet no significant differences were observed (Figure S6). This limitation of the fly model is mentioned in the results, noting the observation of the axonal degeneration phenotype but not alterations in ROS signaling, autophagy activity, or mitochondrial quantity as follows (line 287–290):

“We investigated the impacts of dominant negative mutations on mitochondrial oxidation levels, mitochondrial quantity, and autophagy activation levels; however, none of these parameters showed statistical significance (Figure S6).”

The reviewer also inquired about the effects of overexpressing and underexpressing *OPA1* on axonal count and whether these effects are subject to a threshold. In response, we expressed both wild-type and variant forms of human *OPA1* in *Drosophila* in vivo and assessed their protein levels using Western blot analysis. The results showed no significant differences in expression levels between the wild-type and variant forms in the *OPA1* overexpression experiments, suggesting the absence of a variation threshold effect. These findings have been newly documented as quantitative data in Figure 5C. Furthermore, we have included a statement in the results section for Figure 6A, clarifying that overexpression of hOPA1 exhibited no discernible impact, as detailed on lines 274–276.

“The results presented in Figure 5C indicate that there are no significant differences in the expression levels among the variants, suggesting that variations in expression levels do not influence the outcomes.”

(6) Could the authors clarify what exons in Figure 5 are included in their transcript. My understanding is transcript NM_015560.3 contains exon 4,4b but not 5b. According to Song 2007 this transcript produces invariably s-OPA1 as it contains the exon 4b cleavage site. If this is true, this is a critical limitation in this study and in my opinion significantly undermines the likelihood of the proposed explanation of the findings presented in Figure 6. The primarily functional location of OPA1 is at the IMM and l-OPA1 is the primary opa1 isoform probably only that localizes here as the additional AA act as a IMM anchor. Given this is where GTPase likely oligomerizes the expression of s-OPA1 only is unlikely to interact anyway with native protein. I am not aware of any evidence s-OPA1 is involved in oligomerization. Therefore I don't think this method and specifically expression of a hOPA1 transcript which only makes s-OPA1 to be a reliable indicator of dominant negativity/interference with WT protein function. This could be checked by blotting UAS-hOPA1 protein with a OPA1 antibody specific to human OPA1 only and not to dOPA1. There are several available on the market and if the authors see only s-OPA1 then it confirms they are not expressing l-OPA1 with their hOPA1 construct.

As suggested by the reviewer, we performed a Western blot using a human OPA1 antibody to determine if the expressed hOPA1 was producing the l-OPA1 isoform, as shown in band 2 of Figure 5D. The results confirmed the presence of both l-OPA1 and what appears to be s-OPA1 in bands 2 and 4, respectively. These findings are documented in the updated Figure 5D, with a detailed description provided in the manuscript at lines 224-226. Additionally, the NM_015560.3 refers to isoform 1, which includes only exons 4 and 5, excluding exons 4b and 5b. This isoform can express both l-OPA1 and s-OPA1 (refer to Figure 1 in Song et al., J Cell Biol. 2007). We have updated the schematic diagram in the figure to include these exons. The formation of s-OPA1 through cleavage occurs at the OMA1 target site located in exon 5 and the Yme1L target site in exon 5b of *OPA1*. Isoform 1 of *OPA1* is prone to cleavage by OMA1, but a homologous gene for OMA1 does not exist in *Drosophila*. Although a homologous gene for Yme1L is present in *Drosophila*, exon 5b is missing in isoform 1 of *OPA1*, leaving the origin of the smaller band resembling s-OPA1 unclear at this point.

**Reviewer #2 (Public Review):**
The data presented support and extend some previously published data using *Drosophila* as a model to unravel the cellular and genetic basis of human Autosomal dominant optic atrophy (DOA). In human, mutations in OPA1, a mitochondrial dynamin like GTPase (amongst others), are the most common cause for DOA. By using a *Drosophila* loss-of-function mutations, RNAi- mediated knockdown and overexpression, the authors could recapitulate some aspects of the disease phenotype, which could be rescued by the wild-type version of the human gene. Their assays allowed them to distinguish between mutations causing human DOA, affecting the optic system and supposed to be loss-of-function mutations, and those mutations supposed to act as dominant negative, resulting in DOA plus, in which other tissues/organs are affected as well. Based on the lack of information in the Materials and Methods section and in several figure legends, it was not in all cases possible to follow the conclusions of the authors.

We appreciate the reviewer's constructive feedback and the emphasis on enhancing clarity in our manuscript. We recognize the concerns raised about the lack of detailed information in the Materials and Methods section and several figure legends, which may have obscured our conclusions. In response, we have appended the detailed genotypes of the *Drosophila* strains used in each experiment to a supplementary table. Additionally, we realized that the description of 'immunohistochemistry and imaging' was too brief, previously referenced simply as “immunohistochemistry was performed as described previously (Sugie et al., 2017).” We have now expanded this section to include comprehensive methodological details. Furthermore, we have revised the figure legends to provide clearer and more thorough descriptions.

Similarly, how the knowledge gained could help to "inform early treatment decisions in patients with mutations in hOPA1" (line 38) cannot be followed.

To address the reviewer's comments, we have refined our explanation of the clinical relevance of our findings as follows. We believe this revision succinctly articulates the practical application of our research, directly responding to the reviewer’s concerns about linking the study's outcomes to treatment decisions for patients with hOPA1 mutations. By underscoring the model’s value in differential diagnosis and its influence on initiating treatment strategies, we have clarified this connection explicitly, within the constraints of the abstract’s word limit. The revised sentence now reads: "This fly model aids in distinguishing DOA from DOA plus and guides initial *hOPA1* mutation treatment strategies."

**Reviewer #3 (Public Review):**
Nitta et al. establish a fly model of autosomal dominant optic atrophy, of which hundreds of different OPA1 mutations are the cause with wide phenotypic variance. It has long been hypothesized that missense OPA1 mutations affecting the GTPase domain, which are associated with more severe optic atrophy and extra-ophthalmic neurologic conditions such as sensorineural hearing loss (DOA plus), impart their effects through a dominant negative mechanism, but no clear direct evidence for this exists particularly in an animal model. The authors execute a well-designed study to establish their model, demonstrating a clear mitochondrial phenotype with multiple clinical analogs including optic atrophy measured as axonal degeneration. They then show that hOPA1 mitigates optic atrophy with the same efficacy as dOPA1, setting up the utility of their model to test disease-causing hOPA1 variants. Finally, they leverage this model to provide the first direct evidence for a dominant negative mechanism for 2 mutations causing DOA plus by expressing these variants in the background of a full hOPA1 complement.Strengths of the paper include well-motivated objectives and hypotheses, overall solid design and execution, and a generally clear and thorough interpretation of their results. The results technically support their primary conclusions with caveats. The first is that both dOPA1 and hOPA1 fail to fully restore optic axonal integrity, yet the authors fail to acknowledge that this only constitutes a partial rescue, nor do they discuss how this fact might influence our interpretation of their subsequent results.

As the reviewer rightly points out, neither *dOPA1* nor *hOPA1* achieve a complete recovery. Therefore, we acknowledge that this represents only a partial rescue and have added the following explanations regarding this partial rescue in the results and discussion sections.

Result:

Significantly —> partially (lines 207 and 228) Discussion (lines 329–342):

In our study, expressing *dOPA1* in the retinal axons of *dOPA1* mutants resulted in significant rescue, but it did not return to control levels. There are three possible explanations for this result. The first concerns gene expression levels. The Gal4-line used for the rescue experiments may not replicate the expression levels or timing of endogenous *dOPA1*. Considering that the optimal functionality of *dOPA1* may be contingent upon specific gene expression levels, attaining a wild-type-like state necessitates the precise regulation of these expression levels. The second is a non-autonomous issue. Although *dOPA1* gene expression was induced in the retinal axons for the rescue experiments, many retinal axons were homozygous mutants, while other cell types were heterozygous for the *dOPA1* mutation. If there is a non-autonomous effect of dOPA1 in cells other than retinal axons, it might not be possible to restore the wild-type-like state fully. The third potential issue is that only one isoform of *dOPA1* was expressed. In mouse OPA1, to completely restore mitochondrial network shape, an appropriate balance of at least two different isoforms, l-OPA1 and s-OPA1, is required (Del Dotto et al., 2017). This requirement implies that multiple isoforms of *dOPA1* are essential for the dynamic activities of mitochondria.

The second caveat is that their effect sizes are small. Statistically, the results indeed support a dominant negative effect of DOA plus-associated variants, yet the data show a marginal impact on axonal degeneration for these variants. The authors might have considered exploring the impact of these variants on other mitochondrial outcome measures they established earlier on. They might also consider providing some functional context for this marginal difference in axonal optic nerve degeneration.

In response to the reviewer’s comment regarding the modest effect sizes observed, we acknowledge that the magnitude of the reported changes is indeed small. To explore the impact of these variants on additional mitochondrial outcomes as suggested, we employed markers such as MitoSOX, Atg8, and COXII for validation. However, we could not detect any significant effects of the DOA plus-associated variants using these methods. We apologize for the redundancy, but to address Reviewer #1's fifth question, we present experimental results showing that while the increased ROS signals observed upon *dOPA1* knockdown were rescued by expressing human *OPA1*, the mutant variants 2708-2711del, D438V, and R445H did not ameliorate this effect. This outcome mirrors the axonal degeneration phenotype and is documented in Figure 5F. The following text has been added to the results section lines 248–254:

“Furthermore, we assessed the potential for rescuing ROS signals. Similar to its effect on axonal degeneration, wild-type hOPA1 effectively mitigated the phenotype, whereas the 2708-2711del, D438V, and R445H mutants did not (Figure 5F). Importantly, the I382M variant also reduced ROS levels comparably to the wild type. These findings demonstrate that both axonal degeneration and the increase in ROS caused by dOPA1 downregulation can be effectively counteracted by hOPA1. Although I382M retains partial functionality, it acts as a relatively weak hypomorph in this experimental setup.”

Moreover, utilizing mito-QC, we observed elevated mitophagy in our *Drosophila* model, with these results now included in Figure 2D–H. Given the complexity of the genetics involved and the challenges in establishing lines, autophagy activity was quantified by comparing the ratio of Atg8-1 to Atg8-2 via Western blot analysis. However, no significant alterations were detected across any of the genotypes. Additionally, mitochondrial protein levels derived from COXII confirmed consistent mitochondrial quantities, showing no considerable variance following knockdown. These insights affirm that retinal axon degeneration and mitophagy activation are present in the *Drosophila* DOA model, although the Western blot analysis revealed no significant changes in autophagy activation. Such findings necessitate caution as this model may not fully replicate the molecular pathology of the corresponding human disease. These Western blot findings are presented in Figure S4, with the following addition made to section lines 255–263 of the results:

“We also conducted Western blot analyses using anti-COXII and anti-Atg8a antibodies to assess changes in mitochondrial quantity and autophagy activity following the knockdown of *dOPA1*. Mitochondrial protein levels, indicated by COXII quantification, were evaluated to verify mitochondrial content, and the ratio of Atg8a-1 to Atg8a-2 was used to measure autophagy activation. For these experiments, *Tub-Gal4* was employed to systemically knockdown *dOPA1*. Considering the lethality of a whole-body *dOPA1* knockdown, *Tub-Gal80TS* was utilized to repress the knockdown until eclosion by maintaining the flies at 20°C. After eclosion, we increased the temperature to 29°C for two weeks to induce the knockdown or expression of *hOPA1* variants. The results revealed no significant differences across the genotypes tested (Figure S4A–D).”

In assessing the effects of dominant negative mutations, measurements including ROS levels, the ratio of Atg8-1 to Atg8-2, and the quantity of COXII protein were conducted, yet no significant differences were observed (Figure S6). This limitation of the fly model is mentioned in the results, noting the observation of the axonal degeneration phenotype but not alterations in ROS signaling, autophagy activity, or mitochondrial quantity as follows (line 287–290):

“We investigated the impacts of dominant negative mutations on mitochondrial oxidation levels, mitochondrial quantity, and autophagy activation levels; however, none of these parameters showed statistical significance (Figure S6).”

Despite these caveats, the authors provide the first animal model of DOA that also allows for rapid assessment and mechanistic testing of suspected OPA1 variants. The impact of this work in providing the first direct evidence of a dominant negative mechanism is under-stated considering how important this question is in development of genetic treatments for DOA. The authors discuss important points regarding the potential utility of this model in clinical science. Comments on the potential use of this model to investigate variants of unknown significance in clinical diagnosis requires further discussion of whether there is indeed precedent for this in other genetic conditions (since the model is nevertheless so evolutionarily removed from humans).

As suggested by the reviewer, we have expanded the discussion in our study to emphasize in greater detail the significance of the fruit fly model and the MeDUsA software we have developed, elaborating on the model's potential applications in clinical science and its precedents in other genetic disorders. Our text is as follows (lines 299–318):

“We have previously utilized MeDUsA to quantify axonal degeneration, applying this methodology extensively to various neurological disorders. The robust adaptability of this experimental system is demonstrated by its application in exploring a wide spectrum of genetic mutations associated with neurological conditions, highlighting its broad utility in neurogenetic research. We identified a novel de novo variant in Spliceosome Associated Factor 1, Recruiter of U4/U6.U5 Tri-SnRNP (*SART1*). The patient, born at 37 weeks with a birth weight of 2934g, exhibited significant developmental delays, including an inability to support head movement at 7 months, reliance on tube feeding, unresponsiveness to visual stimuli, and development of infantile spasms with hypsarrhythmia, as evidenced by EEG findings. Profound hearing loss and brain atrophy were confirmed through MRI imaging. To assess the functional impact of this novel human gene variant, we engineered transgenic *Drosophila* lines expressing both wild type and mutant *SART1* under the control of a UAS promoter.

Our MeDUsA analysis suggested that the variant may confer a gain-of-toxic-function (Nitta et al., 2023). Moreover, we identified heterozygous loss-of-function mutations in *DHX9* as potentially causative for a newly characterized neurodevelopmental disorder. We further investigated the pathogenic potential of a novel heterozygous de novo missense mutation in *DHX9* in a patient presenting with short stature, intellectual disability, and myocardial compaction. Our findings indicated a loss of function in the G414R and R1052Q variants of *DHX9* (Yamada et al., 2023). This experimental framework has been instrumental in elucidating the impact of gene mutations, enhancing our ability to diagnose how novel variants influence gene function.”

**Recommendations for the Authors:**

**Reviewer #1 (Recommendations For The Authors):**
Overall I enjoyed reading this paper. It is well presented and represents a significant amount of well executed study. I feel it further characterizes a poorly understood model of OPA1 variants and one which displays significant differences with the human phenotype. However I feel the use of this model with the author's experiments are not enough to validate this model/experiment as a screening tool for dominant negativity. I have therefore suggested the above experiments as a way to both further validate the mitochondrial dysfunction in this model and to ensure that the expressed transcript is able affect oligomerization as this is a pre-requisite to the authors conclusions.

We assessed the extent to which our model reflects mitochondrial dysfunction using COXII, Atg8, and MitoSOX markers. Unfortunately, neither COXII levels nor the ratio of Atg8a-1 to Atg8a-2 showed significant variations across genotypes that would clarify the impact of dominant negative mutations. Nonetheless, MitoSOX and mito-QC results revealed that mitochondrial ROS levels and mitophagy are increased in *Drosophila* following intrinsic knockdown of dOPA1. These findings are documented in Figures 2, 5, and S6.

Regarding oligomer formation, the specifics remain elusive in this study. However, the expression of *dOPA1K273A*, identified as a dominant negative variant in *Drosophila*, significantly disrupted retinal axon organization, as detailed in Figure S7. From these observations, we hypothesize that oligomerization of wild-type and dominant negative forms in *Drosophila* results in axonal degeneration. Conversely, co-expression of *Drosophila* wild-type with human dominant negative forms does not induce degeneration, suggesting that they likely do not interact.

**Reviewer #2 (Recommendations For The Authors):**
Materials and Methods:The authors used GMR-Gal4 to express OPA1-RNAi. (I) GMR is expressed in most cells in the developing eye behind the morphogenetic furrow. So the defects observed can be due to knock- down in support cells rather than in photoreceptor cells.

We have added the following sentences in the result (lines 194–196)."The *GMR-Gal4* driver does not exclusively target Gal4 expression to photoreceptor cells. Consequently, the observed retinal axonal degeneration could potentially be secondary to abnormalities in support cells external to the photoreceptors.”

OPA1-RNAi: how complete is the knock-down? Have the authors tested more than one RNAi line?

We conducted experiments with an additional RNAi line, and similarly observed degeneration in the retinal axons (Figure S2 A and B; lines 178–179).

The loss-of-function allele, induced by a P-element insertion, gives several eye phenotypes when heterozygous (Yarosh et al., 2008). Does RNAi expression lead to the same phenotypes?

A previous report indicated that the compound eyes of homozygous mutations of *dOPA1* displayed a glossy eye phenotype (Yarosh et al., 2008). Upon knocking down *dOPA1* using the *GMR-Gal4* driver, we also observed a glossy eye-like rough eye phenotype in the compound eyes. These findings have been added to Figure S3 and lines 192–194.

There is no description on the way the somatic clones were generated. How were mutant cells in clones distinguished from wild-type cells (e. g. in Fig. 4).

In the Methods section, we described the procedure for generating clones and their genotypes as follows (lines 502–505): "The *dOPA1* clone analysis was performed by inducing flippase expression in the eyes using either *ey-Gal4* with *UAS-flp* or *ey3.5-flp*, followed by recombination at the chromosomal location FRT42D to generate a mosaic of cells homozygous for *dOPA1s3475*." Furthermore, we have created a table detailing these genotypes. In these experiments, it was not possible to differentiate between the clone and WT cells. Accordingly, we have noted in the Results section (lines 201–203): "Note that the mutant clone analysis was conducted in a context where mutant and heterozygous cells coexist as a mosaic, and it was not possible to distinguish between them.”

Why were flies kept at 29{degree sign}C? this is rather unusual.

Increased temperature was demonstrated to induce elevated expression of GAL4 (Kramer and Staveley, Genet. Mol. Res., 2003), which in turn led to an enhanced expression of the target genes. Therefore, experiments involving knockdown assays or Western blotting to detect human OPA1 protein were exclusively conducted at 29°C. However, all other experiments were performed at 25°C, as described in the methods sections: “Flies were maintained at 25°C on standard fly food. For knockdown experiments (Figures 1C–E, 1F–H, 2A–H, 3B–K, 5F, S1, S2 A and B, and S6A), flies were kept at 29°C in darkness.” Furthermore, “We regulated protein expression temporally across the whole body using the *Tub-Gal4* and *Tub-GAL80TS* system. Flies harboring each *hOPA1* variant were maintained at a permissive temperature of 20°C, and upon emergence, females were transferred to a restrictive temperature of 29°C for subsequent experiments.”

Legends:It would be helpful to have a description of the genotypes of the flies used in the different experiments. This could also be included as a table.

We have created a table detailing the genotypes. Additionally, in the legend, we have included a note to consult the supplementary table for genotypes.

Results:Line 141: It is not clear what they mean by "degradation", is it axonal degeneration? And if so, what is the argument for this here?

In the manuscript, we addressed the potential for mitochondrial degradation; however, recognizing that the expression was ambiguous, the following sentence has been omitted: "Nevertheless, the degradation resulting from mitochondrial fragmentation may have decreased the mitochondrial signal.”

Fig. 2: Axons of which photoreceptors are shown?

We have added "a set of the R7/8 retinal axons" to the legend of Figure 2.

Line 167: The authors write that axonal degeneration is more severe after seven days than after eclosion. Is this effect light-dependent? The same question concerns the disappearance of the rhabdomere (Fig. 3G–J).

We conducted the experiments in darkness, ensuring that the observed degeneration is not light- dependent. This condition has been added to the methods section to clarify the experimental conditions.

Line 178/179: Based on what results do they conclude that there is degeneration of the "terminals" of the axons?

Quantification via MeDUsA has enabled us to count the number of axonal terminals, and a noted decrease has led us to conclude axonal terminal degeneration. We have published two papers on these findings. We have added the following description to the results section to clarify how we defined degeneration (lines 174–176): "We have assessed the extent of their reduction from the total axonal terminal count, thereby determining the degree of axonal terminal degeneration (Richard JNS 2022; Nitta HMG 2023).

Line 189: They write: ".. we observed dOPA1 mutant axons...". How did they distinguish es mutant from the controls?Fig. 5 and Fig. 6: How did they distinguish genetically mutant cells from genetically control cells in the somatic clones?

Mutant clone analysis was conducted in a context where mutant and heterozygous cells coexist as a mosaic, and it was not possible to distinguish between them. Accordingly, this point has been added to lines 201–203, “Note that the mutant clone analysis was conducted in a context where mutant and heterozygous cells coexist as a mosaic, and it was not possible to distinguish between them.” and the text in the results section has been modified as follows:

(Before “To determine if *dOPA1* is responsible for axon neurodegeneration, we observed the *dOPA1* mutant axons by expressing full- length versions of *dOPA1* in the photoreceptors at one day after eclosion and found that dOPA1 expression significantly rescued the axonal degeneration”) —>

(After “To determine if *dOPA1* is responsible for axon neurodegeneration, we quantify the number of the axons in the dOPA1 eye clone fly with the expression of *dOPA1* at one day after eclosion and found that dOPA1 expression partially rescued the axonal degeneration”)

Line 225/226: It is not clear to me how their approach "can quantitatively measure the degree of LOF".

To address the reviewer's question and clarify how our approach quantitatively measures the degree of loss of function (LOF), we revised the statement (lines 238–247):

"Our methodology distinctively facilitates the quantitative evaluation of LOF severity by comparing the rescue capabilities of various mutations. Notably, the 2708-2711del and I382M mutations demonstrated only partial rescue, indicative of a hypomorphic effect with residual activity. In contrast, the D438V and R445H mutations failed to show significant rescue, suggesting a more profound LOF. The correlation between the partial rescue by the 2708-2711del and I382M mutations and their classification as hypomorphic is significant. Moreover, the observed differences in rescue efficacy correspond to the clinical severities associated with these mutations, namely in DOA and DOA plus disorders. Thus, our results substantiate the model’s ability to quantitatively discriminate among mutations based on their impact on protein functionality, providing an insightful measure of LOF magnitude.”

Discussion:Line 251, 252 and line 358: What is "the optic nerve" in the adult *Drosophila*?

In humans, the axons of retinal ganglion cells (RGCs) are referred to as the optic nerve, and we posit that the retinal axons in flies are similar to this structure. In the introduction section, where it is described that the visual systems of flies and humans bear resemblance, we have appended the following definition (lines 107–108): “In this study, we defined the retinal axons of *Drosophila* as analogous to the human optic nerve.”

Line 344: These bands appear only upon overexpression of the hOPA1 constructs, so this part of the is very speculative.

Confirmation was achieved using anti-hOPA1, demonstrating that myc is not nonspecific. These results have been added to Figure 5D. Furthermore, the phrase “The upper band was expected as” has been revised to “From a size perspective, the upper band was inferred to represent the full-length hOPA1 including the mitochondria import sequence (MIS).” (lines 464–465)

I was missing a discussion about the increase of ROS upon loss/reduction of dOPA1 observed by others and described here. Is there an increase of ROS upon expression of any of the constructs used?

We demonstrated that not only axonal degeneration but also ROS can be suppressed by expressing human *OPA1* in the genetic background of *dOPA1* knockdown. Additionally, rescue was not possible with any variants except for I382M. Furthermore, we assessed whether there were changes in ROS in the evaluation of dominant negatives, but no significant differences were observed in this experimental system. These findings have been added to the discussion section as follows (lines 318–328). “Our research established that *dOPA1* knockdown precipitates axonal degeneration and elevates ROS signals in retinal axons. Expression of human *OPA1* within this context effectively mitigated both phenomena; it partially reversed axonal degeneration and nearly completely normalized ROS levels. These results imply that factors other than increased ROS may drive the axonal degeneration observed post-knockdown. Furthermore, while differences between the impacts of DN mutations and loss-of- function mutations were evident in axonal degeneration, they were less apparent when using ROS as a biomarker. The extensive use of transgenes in our experiments might have mitigated the knockdown effects. In a systemic *dOPA1* knockdown, assessments of mitochondrial quantity and autophagy activity revealed no significant changes, suggesting that the cellular consequences of reduced *OPA1* expression might vary across different cell types.”

**Reviewer #3 (Recommendations For The Authors):**
Consider being more explicit regarding literature that has or has failed to test a direct dominant negative effect by expressing a variant in question in the background of a full OPA1 complement. My understanding is that this is the first direct evidence of this widely held hypothesis. This lends to the main claim promoting the utility of fly as a model in general. The authors might also outline this in the introduction as a knowledge gap they fill through this study.

In the introduction, we have incorporated a passage that highlights precedents capable of distinguishing between LOF and DN effects, and we note the absence of models capable of dissecting these distinctions within an in vivo organism. This study aims to address this gap, proposing a model that elucidates the differential impacts of LOF and DN within the context of a living model organism, thereby contributing to a deeper understanding of their roles in disease pathology. We added the following sentences in the introduction (lines 71–80).

“In the quest to differentiate between LOF and DN effects within the context of genetic mutations, precedents exist in simpler systems such as yeast and human fibroblasts. These models have provided valuable insights into the conserved functions of OPA1 across species, as evidenced by studies in yeast models (Del Dotto et al., 2018) and fibroblasts derived from patients harboring *OPA1* mutations (Kane et al., 2017). However, the ability to distinguish between LOF and DN effects in an in vivo model organism, particularly at the structural level of retinal axon degeneration, has remained elusive. This gap underscores the necessity for a more complex model that not only facilitates molecular analysis but also enables the examination of structural changes in axons and mitochondria, akin to those observed in the actual disease state.”

The authors should clarify the language used in the abstract and introduction on the effect of hOPA1 DOA and DOA plus on the dOPA1- phenotype. Currently written as "none of the previously reports mutations known to cause DOA or DOA plus were rescued, their functions seems to be impaired." but presumably the authors mean that these variants failed to rescue to the dOPA1 deficient phenotype.

We thank the reviewer for the constructive feedback. We acknowledge the need for clarity in our description of the effects of *hOPA1* DOA and DOA plus mutations on the *dOPA1*- phenotype in both the abstract and the introduction. The current phrasing, "none of the previously reported mutations known to cause DOA or DOA plus were rescued, their functions seem to be impaired," may indeed be confusing. To address your concern, we have revised this statement to more accurately reflect our findings: "Previously reported mutations failed to rescue the *dOPA1* deficiency phenotype." For Abstract site, we have changed as following. "we could not rescue any previously reported mutations known to cause either DOA or DOA plus.”→ “mutations previously identified did not ameliorate the *dOPA1* deficiency phenotype.”

DOA plus is associated with a multiple sclerosis-like illness; as written it suggests that the pathogenesis of sporadic multiple sclerosis and that associated with DOA plus share and underlying pathogenic mechanism. Please use the qualifier "-like illness."

We have added the term “multiple sclerosis-like illness” wherever “multiple sclerosis” is mentioned.